# *MEN1* deficiency leads to neuroendocrine differentiation of lung cancer and disrupts the DNA damage response

Huan Qiu[1,5], Bang-Ming Jin[1,5], Zhan-Feng Wang[2], Bin Xu[1], Qi-Fan Zheng[1], Li Zhang[1], Ling-Yu Zhu[1], Shuang Shi[1], Jun-Bo Yuan[1], Xiao Lin[1], Shu-Bin Gao[1] & Guang-Hui Jin[1,3,4 ✉]

The *MEN1* gene, a tumor suppressor gene that encodes the protein menin, is mutated at high frequencies in neuroendocrine (NE) tumors; however, the biological importance of this gene in NE-type lung cancer in vivo remains unclear. Here, we established an ATII-specific $Kras^{G12D/+}/Men1^{-/-}$ driven genetically engineered mouse model and show that deficiency of menin results in the accumulation of DNA damage and antagonizes oncogenic *Kras*-induced senescence and the epithelial-to-mesenchymal transition during lung tumorigenesis. The loss of menin expression in certain human primary lung cancers correlates with elevated NE profiles and reduced overall survival.

[1] Department of Basic Medical Sciences, School of Medicine, Xiamen University, Xiamen 361102 Fujian Province, People's Republic of China. [2] Department of Neurosurgery, China-Japan Union Hospital of Jilin University, Changchun 130033, People's Republic of China. [3] State Key Laboratory of Cellular Stress Biology, Xiamen University, Xiamen 361102 Fujian Province, People's Republic of China. [4] Cancer Research Center, Xiamen University, Xiamen 361102 Fujian Province, People's Republic of China. [5] These authors contributed equally Huan Qiu, Bang-Ming Jin. ✉email: ghjin@xmu.edu.cn

Lung cancer is the most common cause of cancer-related death worldwide and is classified into two broad histological subtypes: non-small-cell lung cancer (NSCLC) and small-cell lung cancer (SCLC). Clinically, apart from the conventional histological classifications, some types of lung cancer cells possess neuroendocrine (NE) differentiation characteristics; tumors including these cells are called pulmonary NE tumors (P-NETs)[1]. P-NETs comprise a spectrum of tumors including low-grade typical carcinoids (TCs), intermediate-grade atypical carcinoids (ACs), highly malignant SCLCs and large-cell NE carcinoma[2]. The common characteristics of P-NETs are that they possesses NE morphology and express NE markers such as neural cell adhesion protein 1 (NCAM1), neuron-specific enolase (NSE), chromogranin A (CgA), and synaptophysin (Syn)[3].

Lung cancer is histologically classified as a heterogeneous group based on the genetic lesions involved. Retinoblastoma (Rb1) is a critical negative regulator of NE differentiation, and inactivation of *Rb1* frequently occurs in NE-type SCLC[4]. Conversely, *Kras*-activating mutations are mainly found in NSCLC, whereas *p53* mutation or inactivation is broadly associated with both SCLC and NSCLC[5]. *Kras*-activating mutation-driven genetically engineered mouse models (GEMMs) accurately reflect the biology of NSCLC with a mesenchymal profile, whereas *Rb/p53* deletion produces tumors with NE-type SCLC features[6,7].

Clinically, SCLC is designated as a recalcitrant cancer, and targeted agents have failed to demonstrate efficacy to date. In fact, there have been no important advances in clinical therapeutics for this disease for 30 years[8]. The lysine demethylase 1 (LSD1) is active on monomethylated (me1) or dimethylated (me2) histone 3 lysine 9 (H3K9) and H3K4, hence eliminating the obligatory intermediates for the trimethylation (me3) step[9,10]. Recent studies have found that LSD1 inhibitors have excellent physicochemical properties that demonstrate efficacy in SCLC models[9]. These findings indicate a potential role for chromatin remodeling in controlling the development of P-NETs and further suggest that chromatin modifications may serve as therapeutic targets.

The epigenetic regulator menin, the product of the *MEN1* gene (*Men1* in mice), is associated with the inherited tumor syndrome multiple endocrine neoplasia type 1 (MEN1)[11]. In human pancreatic NE tumors (PanNETs), the *MEN1* gene undergoes somatic inactivating mutations with high frequency[12]. In mouse models, loss of *Men1* leads to lethal multiple endocrine tumors[13]. Mechanistically, menin interacts with the mixed lineage leukemia proteins (MLLs), which are orthologues of the trithorax group (TrxG) proteins, and the MLL SET domain specifically catalyzes inherent H3K4me3, leading to activation of cyclin-dependent kinase inhibitor transcription in endocrine organs[14]. *MEN1* deficiency is also associated with lung, gastrointestinal stromal, and prostate cancers[15–18]. Importantly, the *MEN1/MLL* gene is mutated at high frequencies in certain types of human NE lung cancers, such as lung carcinoids or SCLC[15,16], suggesting that menin regulation is a common characteristic in NE-type neoplasms originating from multiple organs, including NE-type lung cancer.

Here, we generated several GEMMs to examine the tumor suppressor activity of menin in NE-type lung tumorigenesis.

## Results

**Loss of *Men1* results in NE-type lung cancer.** Given that homozygous loss of exons 3–8 of the *Men1* gene (*Men1*[−/−]) causes embryonic lethality in mice[13], mice with heterozygous loss of *Men1* (*Men1*[+/−]) were used in the present study. Some of the *Men1*[+/−] mice exhibited labored breathing at ~1 year of age and were killed at 12–18 months. Morphological observation indicated that definite tumors (>1 mm²) and some large tumors (>10

mm²) developed in the central and peripheral lungs of 75% of the *Men1*[+/−] mice (Fig. 1a). Tumor heterogeneity was frequently observed upon hematoxylin and eosin (H&E) staining, and different types of carcinoma were visible even in a single lung, including SCLC, carcinoid cancer, squamous cell carcinoma, and adenocarcinoma (Fig. 1b). The tumor cells showed small, densely hyperchromatic nuclei and poorly developed cytoplasm, and most of them grew in sheets and spread invasively. Immunohistochemistry (IHC) showed that menin staining was reduced and that NE markers, including NCAM1, NSE, Syn, and CgA, were highly stained in the lung tumor area (Fig. 1c, d), confirming the NE features of lung tumors upon *Men1* loss.

Next, we utilized conditional UBC-Cre *MLL* (*MLL*[Δ/Δ]) or *Men1* (*Men1*[Δ/Δ]) allele homozygous knockout (*KO*) mouse models to elucidate whether MLL participates in menin-regulated NE differentiation. *MLL*[Δ/Δ] mice displayed severe symptoms, such as low body weight, weakness, and energielos, and began to die at 3 months after intraperitoneal (i.p.) injection of tamoxifen (TAM). *MLL*[Δ/Δ] mice were killed and broadening of the alveolar septum was observed upon H&E staining; however, no obvious tumors were found in the lung or other organs (Supplementary Fig. 1a). Importantly, *MLL*[Δ/Δ] resulted in upregulation of NCAM1, NSE, Syn, and CgA in lung tissue, suggesting that *MLL* is also involved in controlling NE differentiation (Fig. 1g, Supplementary Fig. 1a). *Men1*[Δ/Δ] mice did not show the same severe symptoms as *MLL*[Δ/Δ] mice; they were viable with normal body weight, although some of them exhibited labored breathing at ~ 8 months. Central and peripheral lung tumors were found in 10% and 43% of mice at 8 and 12 months, respectively (Supplementary Fig. 1b). The lung cancer exhibited strong staining for NCAM1, NSE, Syn, and CgA, and the observed reductions in menin were associated with high Ki67 proliferation indexes in tumors (Fig. 1e, f, Supplementary Fig. 1b, c). Together, these results indicate that *Men1* single-gene deficiency is sufficient for the development of spontaneous mixed-type lung cancer that predominantly exhibits NE differentiation features.

**Men1 deficiency accelerates *Kras*-induced lung tumor.** To rule out the possibility that the lung tumors had metastasized from endocrine organ tumors, lung tissue-specific markers were examined. The IHC results indicated that the lung tumors from *Men1*[+/−] or *Men1*[Δ/Δ] mice were predominantly positive for CGRP (NE cells) and Sftpc (alveolar type II, or ATII cells), and a few tumors were positive for CC10 (Clara cells) (Fig. 2a). The results indicate that *Men1* deficiency gives rise to primary lung cancer and further suggest that there are multiple cellular origins of primary lung cancer attributable to *Men1* deletion. ATII cells are progenitor cells with multidirectional differentiation potential in lung carcinogenesis and serve as the cellular origins of SCLC[6,19]. A mouse model of ATII cell-specific *Kras*[G12D] activation concomitant with *Men1* deletion was developed for the present study (Supplementary Fig. 2a). The *LSL-Kras*[G12D/+]; *Men1*[f/f]; *Sftpc*-Cre (KMS) mice began to exhibit labored breathing and diminished hyperactivity after receiving TAM for 2 weeks. MicroCT imaging revealed that the lung texture was clear, and no abnormalities were observed in wild-type (WT) or *Men1*[f/f]; *Sftpc*-Cre (MS) mice. The lung texture of *LSL-Kras*[G12D/+]; *Sftpc*-Cre (KS) mice was clear and normal at 3 weeks, with slight density at 9 weeks. Strikingly, there was already substantial density at the periphery of KMS lung tissue at 3 weeks, and massive density was observed at up to 9 weeks (Fig. 2b, Supplementary Fig. 2b). The density of KS and KMS lungs developed at similar locations in both distal and peripheral lung tissue. In addition, rapid progression of a donut-like pattern of density in the major bronchi

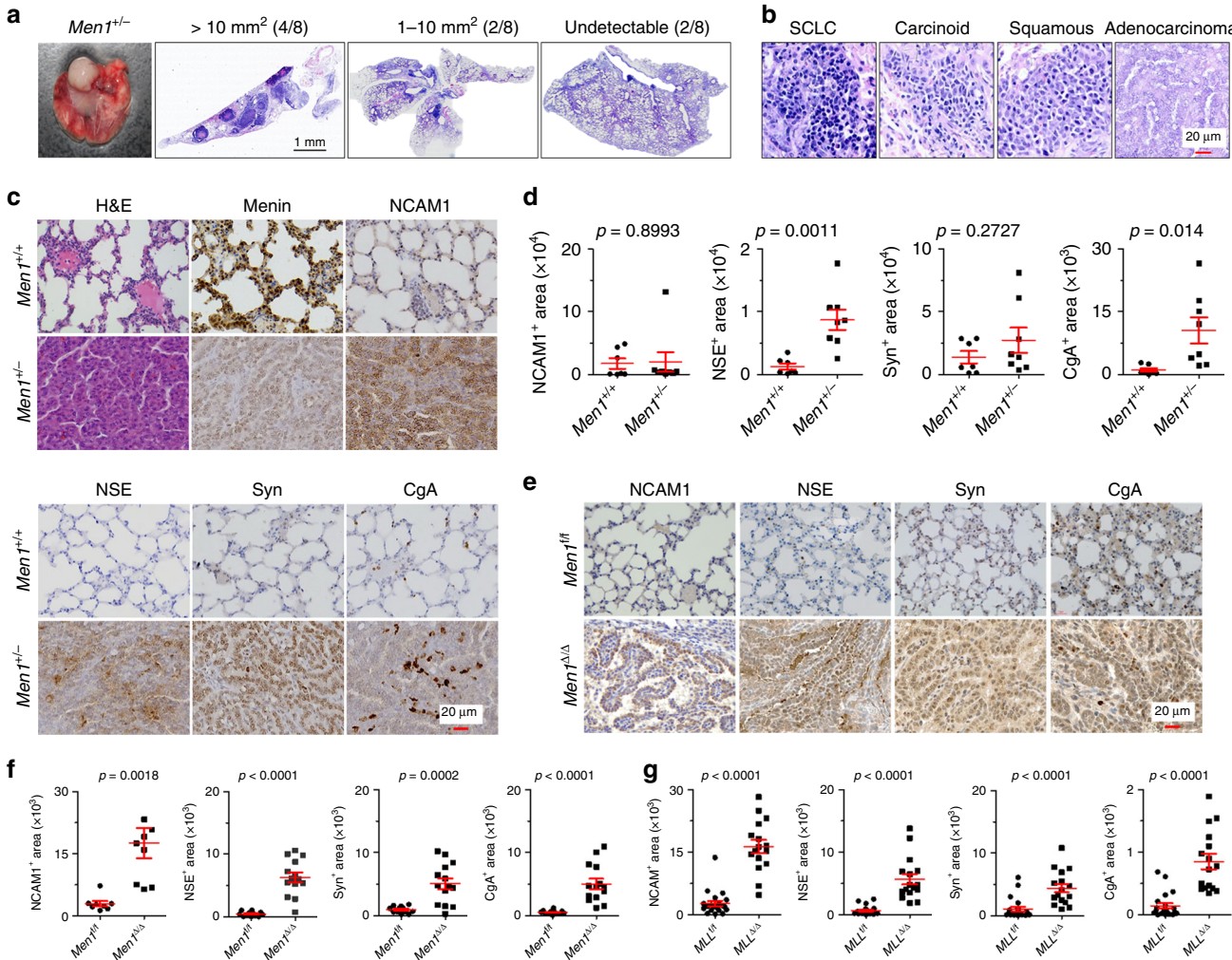

**Fig. 1 Loss of *Men1* induces mixed-type lung cancers with NE profiles. a** Representative brightfield image and H&E images of lungs dissected from *Men1*[+/−] mice at 12–18 months (*n* = 8). The mice are grouped by tumor area: >10 mm$^2$ (4/8), 1–10 mm$^2$ (2/8), or undetectable (2/8). Scale bars, 1 mm. **b** H&E staining of multiple pathological types of lung cancers from *Men1*[+/−] mice, including SCLC, carcinoid cancer, squamous cell carcinoma, and adenocarcinoma. The samples were blindly classified by three independent pathologists. Scale bars, 20 μm. **c** H&E and IHC staining of the indicated antibodies in the lungs of *Men1*[+/+] and *Men1*[+/−] mice. *n* = 8; Scale bars, 20 μm. **d** Automatic quantification of NCAM1, NSE, Syn, CgA IHC staining in *Men1*[+/+] and *Men1*[+/−] 12–18 months mice lung tissues. **e** IHC staining of the indicated antibodies in *Men1*[f/f] (*n* = 12) and *Men1*[Δ/Δ] (*n* = 14) mice at 12 months post intraperitoneal (i.p.) injections of 100 mg/kg TAM. Scale bars, 20 μm. **f** Automatic quantification of NCAM1, NSE, Syn, CgA IHC staining in *Men1*[f/f] (*n* = 12) and *Men1*[Δ/Δ] (*n* = 14) 12 months mice lung tissues. **g** Automatic quantification of NCAM1, NSE, Syn, CgA IHC staining in *MLL*[f/f] (*n* = 21) and *MLL*[Δ/Δ] (*n* = 16) 4 months mice lung tissues, related to Supplementary Fig. 1a. Data are represented as mean ± SEM in **d**, **f** and **g**. Dots in **d**, **f**, and **g** depict individual samples. Significance determined by two-tailed unpaired *t* tests in **d**, **f**, and **g** are indicated. Source Data are provided as a Source Data file.

and large bronchioles was found in KMS lung tissue at 9 weeks (Fig. 2b, Supplementary Fig. 2b); this density obstructed the bronchial/bronchiolar lumen and reduced the alveolar airspace to an extent that ultimately affected gas exchange. The KMS mice became worse over time and began to die from 13 days onward; mice were killed when they became moribund. Morphological observation indicated that KMS lung tissue was filled with tumors, whereas that of KS mice predominantly showed hyperplasia with rare nodes on the surface of the lung (Fig. 2b).

Surprisingly, the survival analysis showed that KMS mice died from 2 weeks onward and entered the stage of rapid death in 1 month, and all KMS mice died within the limit of 80 days (Fig. 2c). The WT and MS mice survived and were healthy throughout the 150 days. The latency period for tumor development in KS mice was between 22 days and 150 days, and most of the KS mice died during the period from 100 to 150 days (Fig. 2c). Remarkably, the lung weight and lung

coefficient (lung weight/body weight) were significantly elevated in KMS mice (Fig. 2d). These findings provide strong evidence that *Men1* deficiency dramatically accelerates oncogenic *Kras* mutation-induced lung tumorigenesis.

H&E staining indicated that lung tumors arising from KMS mice frequently contained several nodules and often consisted of more than one tumor cell type, including SCLC, adenocarcinoma, and mixed types (Supplementary Fig. 2c, d). The tumor cells were positive for Sftpc but negative for CC10 and menin, suggesting that the tumors indeed arose from specific *Men1*-deleted ATII cells (Supplementary Fig. 2d). Consistent with a previous report[20], the tumors in KS mice exhibited low staining for the epithelial marker E-cadherin and high levels of mesenchymal markers (Nestin, Vimentin, and ZEB1), whereas *Men1* deficiency in KMS mice clearly recovered E-cadherin staining and significantly reduced the expression of Nestin, Vimentin, and ZEB1 in lung tumors (Fig. 2e, Supplementary Fig. 2e).

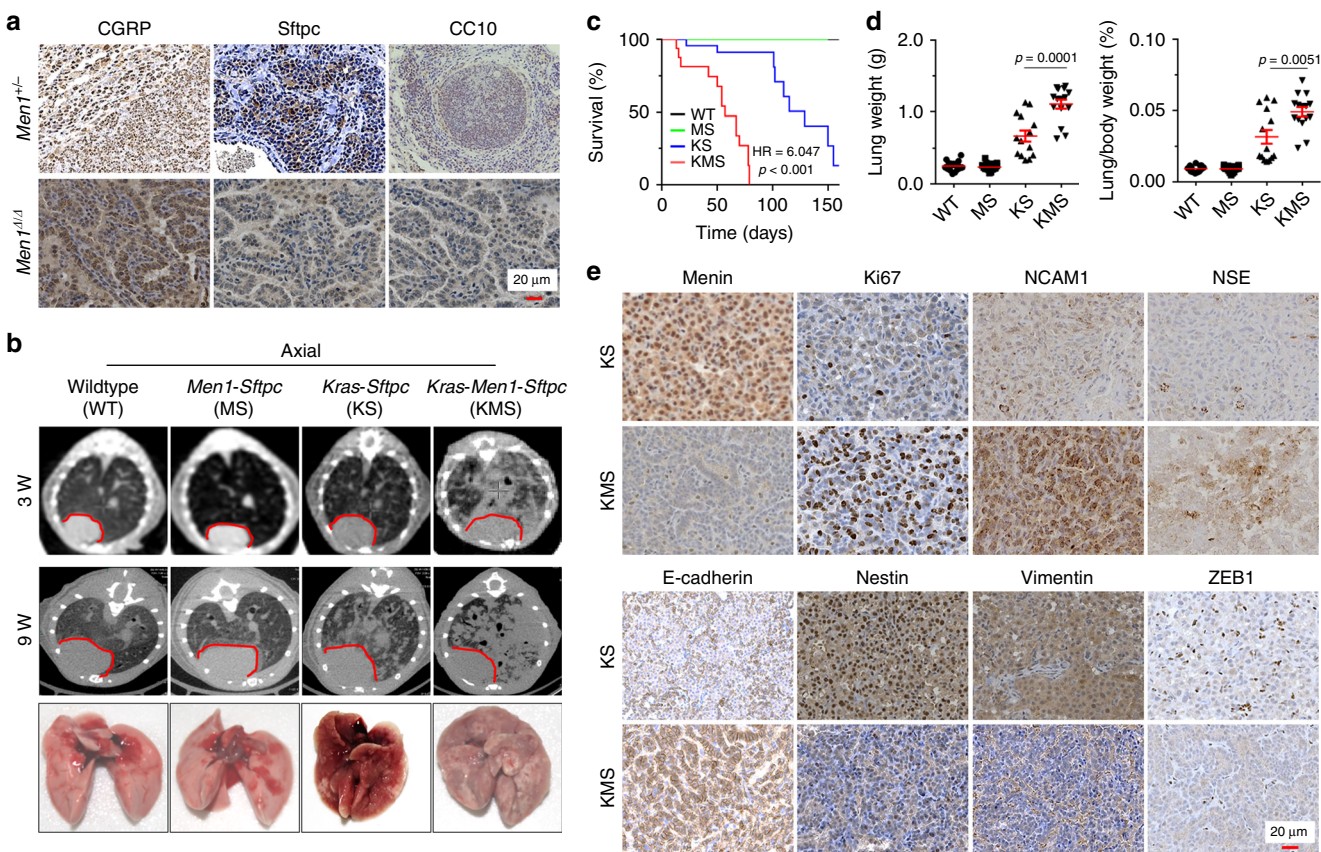

**Fig. 2 *Men1* deficiency accelerates the development of *Kras*-induced lung cancer. a** IHC staining was performed with the indicated specific markers in lung tumors from $Men1^{+/-}$ and $Men1^{\Delta/\Delta}$ mice (from Fig. 1). CGRP, NE cells; Sftpc, ATII cells; CC10, Clara cells. Scale bars, 20 μm. **b** Representative axial thoracic microCT images from mice of the indicated genotypes at 3 and 9 weeks after i.p. injection of TAM. The red line marks the location of the pericardium. The brightfield images show lung tissues from mice of the indicated genotypes at 9 weeks. **c** Kaplan-Meier survival analysis of mice of the indicated genotypes (WT: $n = 17$; MS: $n = 15$; KS: $n = 14$; KMS: $n = 16$). **d** Scatter plot of lung weight and lung weight/body weight of the indicated mice at 9 weeks (WT: $n = 17$; MS: $n = 15$; KS: $n = 14$; KMS: $n = 15$). **e** IHC staining for the indicated proteins in the lung tumors of KS and KMS mice at 9 weeks. Scale bars, 20 μm. Data are represented as mean ± SEM in **d**. Dots in **d** depict individual samples. The hazard ratios (HR) and *p* values by log-rank (Mantel-Cox) test are indicated in **c**. KS mice compared with KMS mice, $p < 0.0001$. Significance determined by two-tailed unpaired *t* tests are indicated in **d**. Source Data are provided as a Source Data file.

Furthermore, loss of menin promoted the acquisition of an NE differentiation phenotype in tumors as revealed by high expression of NE markers, including NCAM1, NSE, CgA, and Syn (Fig. 2e, Supplementary Fig. 2e, f). The tumors showed high proliferation indexes, as determined by Ki67 immunostaining (Fig. 2e, Supplementary Fig. 2e). These findings suggest that menin is required for the *Kras* mutation-induced epithelial-to-mesenchymal transition (EMT) and that menin antagonizes NE differentiation. Staining for F4/80, a macrophage marker, indicated that extensive macrophage recruitment occurred in KS lung tissue, whereas recruitment was notably blocked by *Men1* deficiency (Supplementary Fig. 2f). It suggests that *Men1* deficiency promotes KS tumor evade macrophage immune surveillance. Finally, CD31 staining in KMS lung tumors was clearly elevated (Supplementary Fig. 2 f), suggesting that tumor neovascularization was activated by the loss of menin. The expression of HMGA2, a major factor for tumor invasiveness and metastasis, was also evidently increased in KMS lung tumors (Supplementary Fig. 2f). Our data indicate that *Men1* deficiency markedly accelerates *Kras*-induced lung carcinogenesis and induces the differentiation of epithelial cells into NE cells.

**Loss of menin leads to inactivation of p53/Rb pathways**. The inactivation of the p53/Rb pathway gives rise to SCLC[7]. IHC revealed that $Men1^{+/-}$ clearly decreased p53 and increased p-Rb (the inactivated form of Rb) staining (Fig. 3a). Similar results were also found in lung tissue from MS and KMS mice (Fig. 3b). Genomic *MYC* amplification is frequently associated with the NE-low "variant" subset of SCLC with a NEUROD1$^{high}$/ASCL1$^{low}$ profile[21]. Here, we found that $Men1^{\Delta/\Delta}$ lung tissue exhibited strong ASCL1 staining with low expression of MYC and NEUROD1 (Supplementary Fig. 1c). These findings suggest that menin controls NE differentiation in a MYC pathway-independent manner. We focused on elucidating how menin influences the expression of p53. Both sh*MEN1* and *Men1* deficiency decreased p53 and 53BP1 protein and increased p-Rb and MDM2, but not MDMX, in A549, MEF cells, and lung tissues, respectively (Fig. 3c, Supplementary Fig. 3a). Overexpression of menin also increased the protein levels of p53 and 53BP1, and decreased MDM2 protein in A549 cells (Supplementary Fig. 3b). Conversely, inactivation of p53 did not obviously alter the expression of menin in sh*p53* A549 and $p53^{\Delta/\Delta}$ MEF cells (Supplementary Fig. 3c). In addition, loss of menin did not clearly affect the mRNA levels of either *p53* or *MDM2* (Supplementary Fig. 3d), suggesting that menin regulates p53/MDM2 through a posttranscriptional mechanism. As expected, co-immunoprecipitation (Co-IP) assays indicated that loss of menin markedly elevated endogenous ubiquitination (ubi)-p53

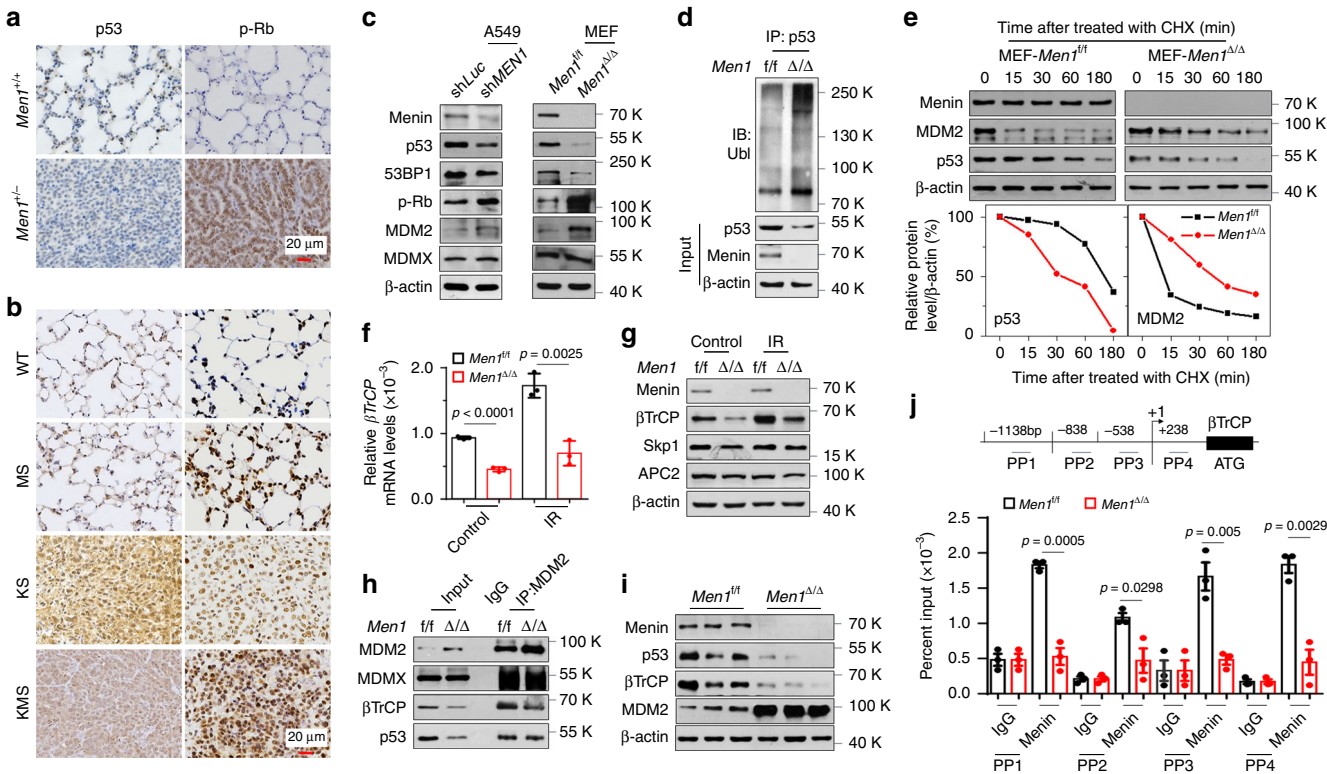

**Fig. 3 Loss of *Men1* leads to inactivation of the p53/Rb pathway. a**, **b** IHC staining was performed with specific p53 and p-Rb$^{Ser780}$ antibodies in lung sections from mice of the indicated genotypes at 18 months **a** and 9 weeks **b**, respectively (from Figs. 1 and 2). The *Men1$^{+/+}$*, WT, and MS were normal lung tissues, the KS and KMS were lung tumors. Scale bars, 20 μm. **c** Western blotting was used to detect the indicated proteins in A549 and MEF cells. **d** Co-IP with an anti-p53 antibody and ubi-p53 detection with an anti-ubiquitin antibody were performed via western blotting of samples from *Men1$^{f/f}$* and *Men1$^{\Delta/\Delta}$* MEF cells treated with 25 μg/ml MG132 for 6 h. **e** *Men1$^{f/f}$* and *Men1$^{\Delta/\Delta}$* MEF cells were treated with 50 μg/ml cyclohexamide (CHX), and the expression of menin, MDM2, and p53 was determined by western blotting (top). The relative expression of p53 or MDM2 was quantified by gray scanning, and the protein levels at the indicated time points are presented relative to those at 0 min (bottom). **f** RT-qPCR was used to detect the mRNA expression of *βTrCP* in *Men1$^{f/f}$* and *Men1$^{\Delta/\Delta}$* MEF cells 1 hour after exposure to 10 Gy of IR. **g** Western blotting was used to detect the expression of the indicated proteins in *Men1$^{f/f}$* and *Men1$^{\Delta/\Delta}$* MEF cells 1 hour after exposure to 10 Gy of IR. **h** Cell lysates were immunoprecipitated with anti-MDM2, and western blotting with MDM2, MDMX, βTrCP, and p53 antibodies was performed for *Men1$^{f/f}$* and *Men1$^{\Delta/\Delta}$* MEF cells. IgG served as the negative control. **i** Western blotting was used to detect the expression of MDM2, βTrCP, and p53 in the lung tissues/tumors of *Men1$^{f/f}$* and *Men1$^{\Delta/\Delta}$* mice (n = 3 per group). **j** Schematic representation of the *βTrCP* gene promoter regions and primer pairs used for ChIP assays. ChIP-qPCR was performed with an anti-menin antibody on samples from *Men1$^{f/f}$* and *Men1$^{\Delta/\Delta}$* MEF cells, and IgG served as the negative control. Data are represented as mean ± SD in **f**, **e** and as mean ± SEM in **j**. Dots in **f** and **j** depict one repeat. Significance determined by two-tailed unpaired *t* tests in **f** and **j** are indicated. Results in **c–i**, and **j** are representative of three independent experiments. Source Data are provided as a Source Data file.

and reduced ubi-MDM2 in *Men1$^{\Delta/\Delta}$* MEF cells (Fig. 3d, Supplementary Fig. 3e). Furthermore, western blotting detection indicated that the half-life of p53 protein was reduced from >60 to 30 min but that the half-life of MDM2 was prolonged from <15 to ~50 min in *Men1$^{\Delta/\Delta}$* MEF cells (Fig. 3e). These findings imply that menin strongly upregulates the expression of p53 through an MDM2-mediated ubiquitination degradation pathway.

Analysis of the amino-acid sequence of menin did not reveal homology to any other known E3 ubiquitin ligase domain, suggesting that menin may regulate the stability of MDM2/p53 indirectly by regulating E3 ligases. We therefore screened a series of E3 ligases and deubiquitination enzymes (DUBs) that have been reported to regulate MDM2 ubiquitination[22]. Real-time quantitative PCR (RT-qPCR) assays confirmed that several E3 ligases, including *βTrCP*, *RBX1*, and *Skp1*, but not DUBs, were markedly downregulated in *Men1$^{\Delta/\Delta}$* MEF cells (Supplementary Fig. 3f). The βTrCP-mediated MDM2 degradation pathway increases p53 activity during the DNA damage response (DDR)[23]. We observed that the expression of *βTrCP* was notably stimulated by exposure to therapeutic ionizing radiation and was further blocked by deletion of *Men1* in MEF cells (Fig. 3f, g). The expression of Skp1

and APC2 was inhibited by *Men1* deletion but was not clearly affected by exposure to infrared (IR) (Fig. 3g). Simultaneously, the interaction between MDM2 and βTrCP, but not that of MDM2 and MDMX, was diminished by deletion of *Men1* in MEF cells (Fig. 3h). Consistently, the expression of βTrCP and p53 was downregulated and that of MDM2 was upregulated in lung tissue from the previously described *Men1$^{\Delta/\Delta}$* GEMMs in vivo (Fig. 3i). The chromatin immunoprecipitation (ChIP) assays clearly indicated that menin binds to the promoter region of *βTrCP* and that this binding is accompanied by abundant H3K4me3 modification, which is notably decreased in *Men1$^{\Delta/\Delta}$* MEF cells (Fig. 3j, Supplementary Fig. 3g). Importantly, we found that upregulation of βTrCP and p53, and reduction of MDM2 by menin overexpression were counteracted by siRNA-mediated knockdown (KD) of βTrCP in A549 cells (Supplementary Fig. 3h). Next, we observed that KD of *MEN1* reduces the expression of Rb and enhances phosphorylation of Rb (p-Rb), whereas the mRNA levels of *Rb* were not remarkably altered in A549 cells (Supplementary Fig. 3i, j). Interestingly, the half-life of p-Rb, but not Rb, was prolonged from <1 to ~12 h in A549-sh*MEN1* cells (Supplementary Fig. 3j). These results is consistent with previous

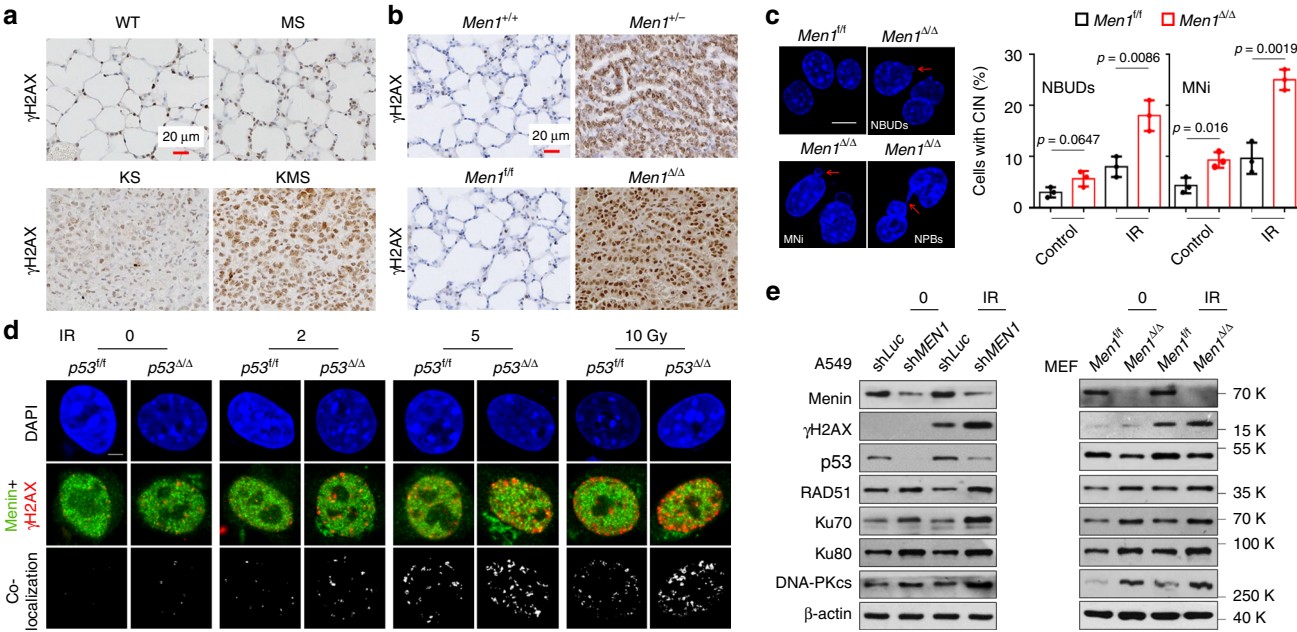

**Fig. 4 Loss of *Men1* triggers aberrant DNA damage repair. a**, **b** IHC staining for γH2AX was performed in lung tissue/tumor sections from mice of the indicated genotypes. **a** The WT, MS, KS, and KMS 11 weeks mice lung tissues. **b** The *Men1*[+/−] 18 months mice lung tissues and the *Men1*[Δ/Δ] 12 months mice lung tissues. Scale bars, 20 μm. **c** Representative images of DAPI staining showing NBUDs, NPBs, and MNi (red arrows) in *Men1*[f/f] and *Men1*[Δ/Δ] MEF cells 1 hour after exposure to 10 Gy of IR. The statistical results were obtained from analysis of mitotic cells (*n* = 100 per group), and repeat the count three times. Scale bars, 10 μm. **d** IF staining for γH2AX (red), menin (green), and DAPI (blue) in *p53*[f/f] and *p53*[Δ/Δ] MEF cells 1 hour after exposure to the indicated IR dose. Colocalization of menin and γH2AX is shown in gray. Scale bars, 10 μm. **e** Western blotting was used to detect the expression of the indicated proteins in A549 and MEF cells 1 hour after exposure to 10 Gy of IR. Data are represented as mean ± SD in **c**. Significance determined by two-tailed unpaired *t* tests in **c** are indicated. Results in **c**, **d**, and **e** are representative of three independent experiments. Source Data are provided as a Source Data file.

report that menin positively regulates Rb activity in a posttranscriptional manner[24]. Altogether, our data demonstrate that menin is a pivotal regulator for p53/Rb pathways.

**Loss of menin triggers aberrant DNA damage response.** Chromosomal instability is one of the distinguishing features of NE-type neoplasms[4]. We were interested in determining whether menin maintains genomic stability in lung epithelial cells. Using the established GEMMs described above, we observed that loss of *Men1* strongly increased DNA damage in lung cancer tissues, as confirmed by increased staining for phosphorylated H2AX (γH2AX) (Fig. 4a, b). Further γH2AX immunofluorescence (IF) staining showed that the DDR was rapidly activated by whole-body exposure to 3 Gy of IR and was further aggravated by *Men1*[Δ/Δ] in lung tissues, even in *Men1*[Δ/Δ] MEF cells (Supplementary Fig. 4a, b, c). The number of γH2AX foci decreased in menin-overexpressing BEAS-2B, A549, and NCI-H446 cells exposed to IR (Supplementary Fig. 4d–g). Moreover, nuclear buds, nucleoplasmic bridges, and micronuclei, markers of CIN[25], were also increased in *Men1*[Δ/Δ] MEF cells exposed to IR (Fig. 4c). These findings indicate that inactivation of menin results in the accumulation of CIN.

We suspect that menin maintains genomic stability in a p53-dependent manner. IF staining showed that the colocalization of menin with γH2AX was increased by exposure to IR in a dose-dependent manner and was further elevated by *p53*[Δ/Δ] (Fig. 4d, Supplementary Fig. 5a). Similar results were obtained in A549 (*p53* WT) and NCI-H1299 (*p53* null) lung cancer cell lines (Supplementary Fig. 5b, c). p53 maintains genomic stability by negatively regulating DNA homologous recombination (HR) repair activity[26]. Consistently, the expression of RAD51, an HR

marker, was notably increased in *p53*[Δ/Δ] MEF cells with or without IR, whereas Ku70 and Ku80, the key mediators of the nonhomologous end-joining (NHEJ) repair pathway, were markedly decreased (Supplementary Fig. 5d). Similarly, loss of menin clearly promoted the expression of RAD51 in sh*MEN1* A549 cells and *Men1*[Δ/Δ] MEF cells with or without IR (Fig. 4e). However, the expression of Ku70/Ku80 and DNA-PKcs was activated by *Men1*[Δ/Δ] with or without IR (Fig. 4e), in contrast to the reduction in Ku70/Ku80 mediated by *p53*[Δ/Δ] (Supplementary Fig. 5d). In addition, ATM and RAD3-related (ATR) phosphorylation (p-ATR) was strongly decreased and p-ATM (ataxia telangiectasia mutated) was increased in *Men1*-deficient A549 and MEF cells after IR treatment (Fig. 4e, Supplementary Fig. 5g). These findings suggest that loss of *Men1* inhibits ATR and that the ATM pathway is activated to compensate in response to DNA damage. Altogether, these findings confirm that menin is a pivotal regulator that maintains genomic stability through p53-dependent HR and p53-independent error-prone NHEJ mechanisms.

Interestingly, we further observed that DNA repair rates (RAD51/γH2AX) were notably decreased in *Men1*[Δ/Δ] relative to *Men1*[f/f] MEF cells (Supplementary Fig. 5e). Loss of *Men1* did not affect the generation of reactive oxygen species (ROS), an inducer of DNA damage, in MEF cells exposure to IR (Supplementary Fig. 5f). Deletion of *Men1* distinctly triggered the phosphorylation of cell cycle checkpoints CHK1 (p-CHK1) and CHK2 (p-CHK2) induced by IR exposure in MEF cells (Supplementary Fig. 5g). Finally, we also observed that exposure to IR markedly activated the expression of p53, p-p53 (p53 phosphorylated on Ser15), and βTrCP and diminished the expression of MDM2 in a dose-dependent manner, effects that were neutralized by the deletion of *Men1* in MEF cells (Supplementary Fig. 5h). Measurement of

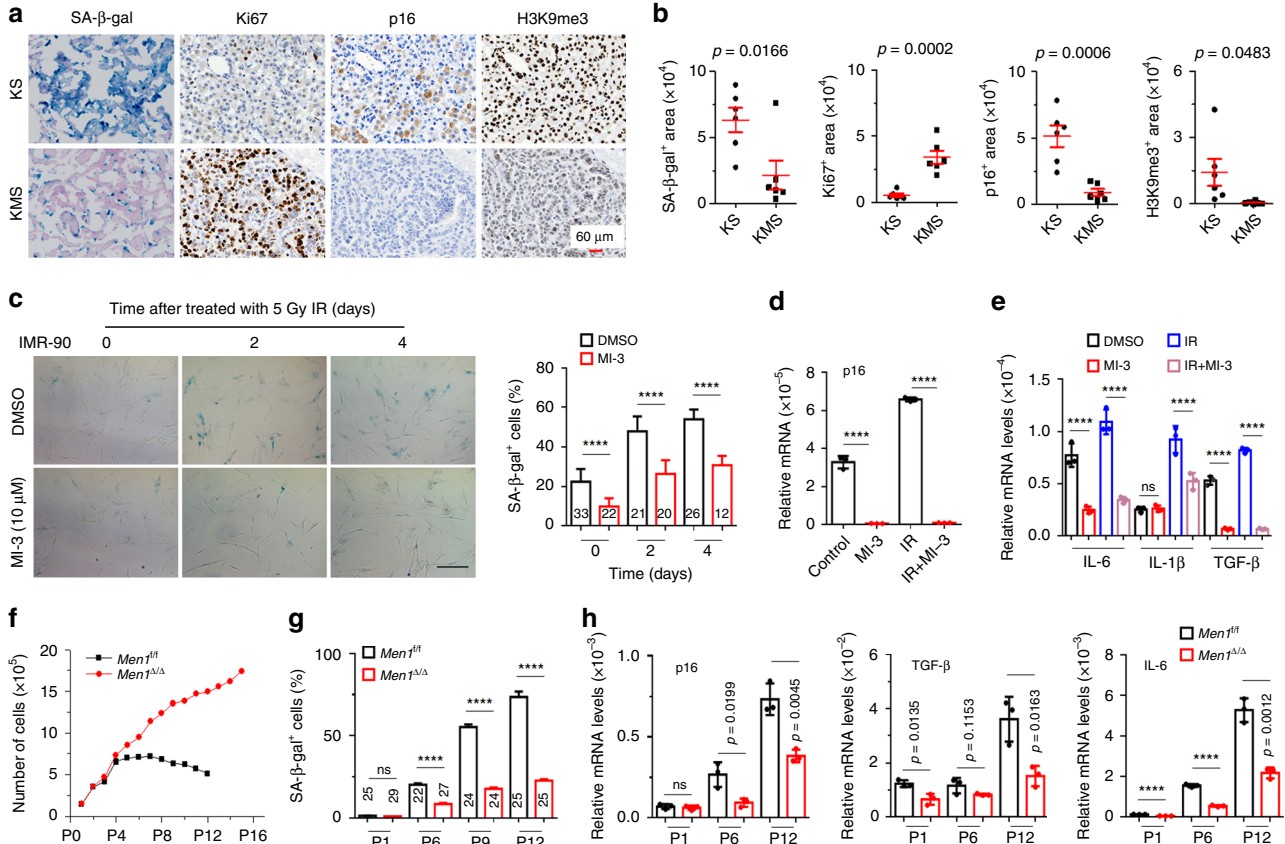

**Fig. 5 Menin is required for the *Kras*-induced OIS programs. a** Representative images of SA-β-gal staining and IHC staining of Ki67, p16, and H3K9me3 in lung consecutive sections from 11 weeks KS or KMS mice. Scale bars, 60 μm. **b** Quantification of SA-β-gal staining, Ki67, p16, and H3K9me3 staining in **a**, $n = 6$ per group. **c** Images of SA-β-gal staining and quantification of SA-β-gal-positive cells. The cells were treated with 10 μm MI-3 for 2 days before receiving 5 Gy of IR, and SA-β-gal staining was performed at the indicated time points after IR exposure. Scale bars, 100 μm. **d**, **e** IMR-90 cells were treated with 10 μm MI-3 for 2 days before IR and were collected 2 days after exposure to 5 Gy of IR. RT-qPCR was used to detect the mRNA levels of *p16*, *IL-6*, *IL-1β*, and *TGF-β*. **f** Growth curve of primary *Men1*f/f and *Men1*Δ/Δ MEF cells during serial passages. Isolation and culture of primary MEF cells were performed as described in the Methods. The cells were treated with 1 μm TAM and seeded in 6 cm dishes every 3 days. $n = 3$ for *Men1*f/f and *Men1*Δ/Δ, respectively. **g** Quantification of the SA-β-gal-positive cells shown in Supplementary Fig. 6i. **h** The mRNA expression of *p16*, *TGF-β*, and *IL-6* was measured by RT-qPCR in primary *Men1*f/f and *Men1*Δ/Δ MEF cells at the indicated passages. Data are represented as mean ± SEM in **b**, **g** and as mean ± SD in **c**, **d**, **e**, and **h**. Dots in **d** depict individual samples and in **d**, **e**, and **h** depict no repeat. Significance determined by two-tailed unpaired *t* tests in **b**, **c**, **d**, **e**, **g**, and **h** are indicated. ****$p < 0.0001$. Results in **c**–**g**, and **h** are representative of three independent experiments. Source Data are provided as a Source Data file.

menin, Skp1, and Cullin1 expression at the indicated times did not reveal clear responses to IR exposure (Supplementary Fig. 5h). These results suggest that menin maintains genomic stability at least partly by controlling cell cycle checkpoints, but not by generation of endogenous ROS.

**Menin is required for oncogenic *Kras*-induced senescence.** We further investigated how menin influences oncogenic *Kras*-induced lung carcinogenesis. It is well known that activation of the *Kras*-mediated oncogene-induced senescence (OIS) program is a fundamental barrier for the initiation of lung cancer[27]. Consistent with this, extensive senescence was confirmed by standard senescence-associated β-galactosidase (SA-β-gal) staining in lung tissues of KS mice; however, loss of *Men1* substantially reduced SA-β-gal staining and increased Ki67 expression in KMS mice (Fig. 5a, b). Furthermore, *p16*[INK4A], a critical OIS tumor suppressor and senescence mediator, and H3K9me3, a marker of senescence-associated heterochromatic foci[28,29], Both of them were significantly repressed by the absence of *Men1* in both KMS and *Men1*Δ/Δ mice (Fig. 5a, b, Supplementary Fig. 6a). The mRNA levels of the standard senescence-associated secretory

phenotype genes, such as *TGF-β* and *IL-6*, were markedly lower in the lung tissues of KMS with loss of *Men1* (Supplementary Fig. 6b). We further established TAM-inducible permanently *Men1* deletion primary MEF cells. SA-β-gal staining indicated that primary *Men1*f/f cells infected with retrovirus particles expressing *Kras*[G12D] underwent senescence, and loss of *Men1* strikingly attenuated OIS (Supplementary Fig. 6c). Moreover, *Men1*Δ/Δ markedly reduced the global levels of chromatin H3K4me3 but not the levels of H3K4me1 or H3K4me2 (Supplementary Fig. 6d). Furthermore, ChIP assays confirmed that menin binds to the promoter region of the *p16* gene, and deletion of *Men1* notably decreased menin binding and H3K4me3 modification (Supplementary Fig. 6e).

To confirm that these results were not unique to *Kras*-induced OIS, we tested a second model of therapy-induced senescence. As expected, exposure to IR resulted in time-dependent induction of cellular senescence and upregulation of *p16* and SASP genes (*IL-6*, *IL-1β*, and *TGF-β*) in IMR-90 cells, but these effects were attenuated by MI-3, a specific inhibitor of menin/MLL interaction (Fig. 5c–e). Notably, markedly decreased H3K4me3 at *p16* promoter regions was observed in IMR-90 cells treated with MI-3 (Supplementary Fig. 6f). Similarly, IR

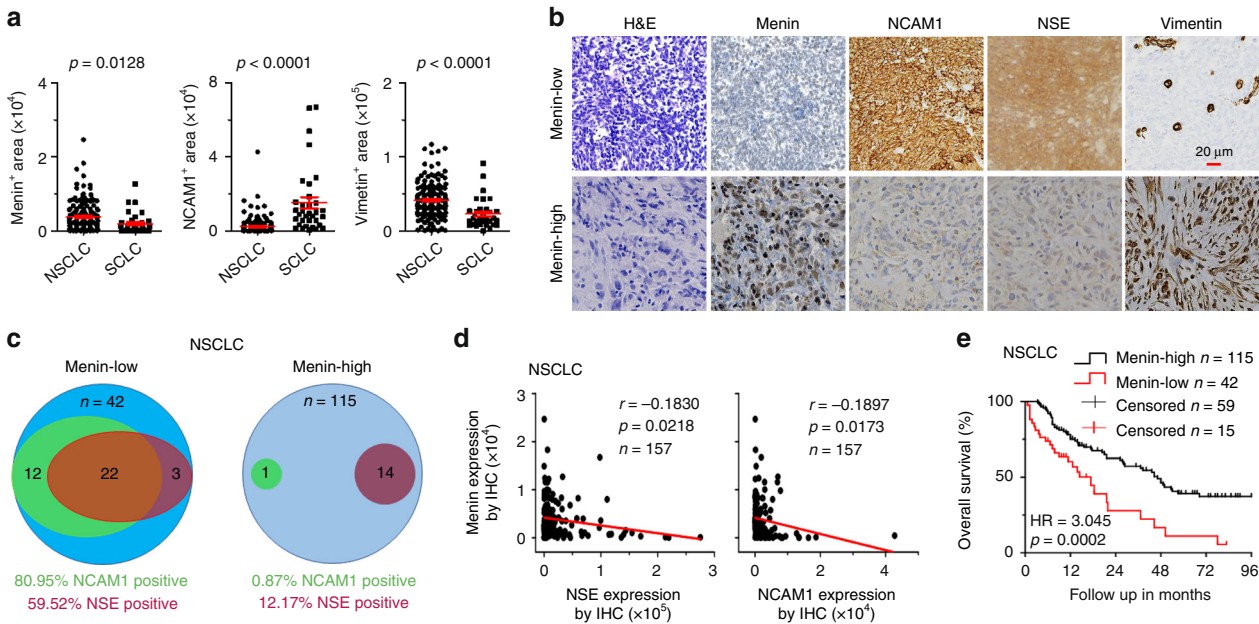

**Fig. 6 Menin expression is correlated with poor prognosis and NE differentiation in human primary lung cancer. a** Automatic quantification of menin, NCAM1, vimentin in human NSCLC ($n = 157$) and SCLC ($n = 34$) specimens IHC staining. **b** H&E and IHC staining for the indicated proteins in clinical lung cancer samples. High or low indicates that the expression of menin in the tumor was similar to or lower than that in tumor-adjacent tissues, respectively. High, menin$^+$ area $\geq$ 1000; low, menin$^+$ area < 1000. Scale bars, 20 μm. **c** Venn diagrams show the number of samples with NCAM1 and/or NSE high expression in menin-low and menin-high NSCLC samples. **d** Correlation analysis between menin and NCAM1 expression, and menin and NSE expression in 157 NSCLC samples. **e** Kaplan-Meier survival analysis for NSCLC according to menin expression (low, $n = 42$; high, $n = 115$; adenocarcinoma: $n = 144$; adenosquamous carcinoma: $n = 13$). Data are represented as mean ± SEM in **a**. Dots in **a**, **d** depict individual samples. Significance determined by two-tailed unpaired $t$ tests in **a** are indicated. The Spearman correlation and $p$ value by Spearman's test are indicated in **d**. The Hazard Ratios (HR) and $p$ values by log-rank (Mantel-Cox) test are indicated in **e**. Source Data are provided as a Source Data file.

exposure evidently increased SA-β-gal staining and enhanced the expression of p16 in BEAS-2B cells, and these effects were further enhanced by *MEN1* overexpression (Supplementary Fig. 6g, h). Normal cells divide until they reach a senescence phase as a result of telomere shortening or stress[30]. Using separated primary MEF cells, we observed that the proliferation capacity of *Men1*$^{f/f}$ MEF cells gradually declined after passage 6 and found that growth was slowed at passage 12 (Fig. 5f). However, *Men1*$^{\Delta/\Delta}$ MEF cells exhibited continuous proliferation in 16 culture passages (Fig. 5f). Among *Men1*$^{f/f}$ MEF cells, the number of SA-β-gal-positive cells increased with successive culture passages, but very few SA-β-gal-positive *Men1*$^{\Delta/\Delta}$ MEF cells were detected (Fig. 5g, Supplementary Fig. 6i), which was negatively correlated with the growth kinetics of these cells (Fig. 5f). The expression of *p16* and SASP genes (*IL-6* and *TGF-β*) gradually increased during culture progression but was blocked in *Men1*$^{\Delta/\Delta}$ MEF cells (Fig. 5h). Altogether, the results demonstrate that menin is fundamentally required for the senescence program and that inactivation of menin in proliferating lung cancer cells enables the cells to bypass *Kras*-induced OIS.

Menin is correlated with poor prognosis and NE lung cancer in order to elucidate whether menin expression correlates with NE differentiation in human NSCLC specimens, we randomly collected human primary lung cancer specimens, including histologically classified specimens from 157 cases of NSCLC and 34 cases of SCLC (Supplementary Table 1), and performed menin IHC staining. Compared with those from NSCLC specimens, samples from SCLC patients possessed markedly strong NE profiles and exhibited reduced mesenchymal characteristics (Fig. 6a, b), consistent with the common idea that NE differentiation is the hallmark of SCLC. The IHC result showed

that the expression of menin in 42 out of 157 NSCLC cases (26.8%) were obviously reduced compared to that in adjacent lung tissues (Fig. 6c). The statistical analysis showed that 81% of cases (34/42) were positive staining for NCAM1, 59.5% of cases (25/42) were positive staining for NSE, and 52.4% of cases (22/42) were positive for both NCAM1 and NSE in menin-low NSCLC samples (Fig. 6c). Conversely, only 0.9% (1/115) cases of NCAM1-positive and 12.2% cases (14/115) of NSE-positive were observed in menin-high NSCLC samples (Fig. 6c). These results indicated that the menin-low patients had significantly high staining for NE markers (NCAM1, NSE) compared with menin-high NSCLC patients (Fig. 6c). Further, the correlation analysis of menin and NE differentiation in 157 NSCLC samples showed that menin is negatively correlated with NCAM1 and NSE expression (Fig. 6d), respectively.

Finally, we performed a correlation analysis between menin expression and malignancy for the 157 cases of NSCLC with detailed clinical data. Our data showed that the low expression of menin was positively correlated with age-of-onset of lung cancer ($p = 0.038$), but there were no significant correlations between menin expression and gender or other clinicopathological characteristics (Supplementary Table 1). Importantly, menin-low patients had shorter median survival times than menin-high patients, with average survival times of 29 and 58.5 months, respectively (Supplementary Table 1). The Kaplan-Meier survival curves were plotted for patients according to their menin expression levels. Strikingly, 9-year overall survival was significantly lower for menin-low patients than for menin-high patients (Fig. 6e). These results further support the conclusion that menin is an essential regulator of lung cancer cell differentiation and demonstrate, for the first time, that low menin expression is a potential predictor of poor outcomes in lung cancer.

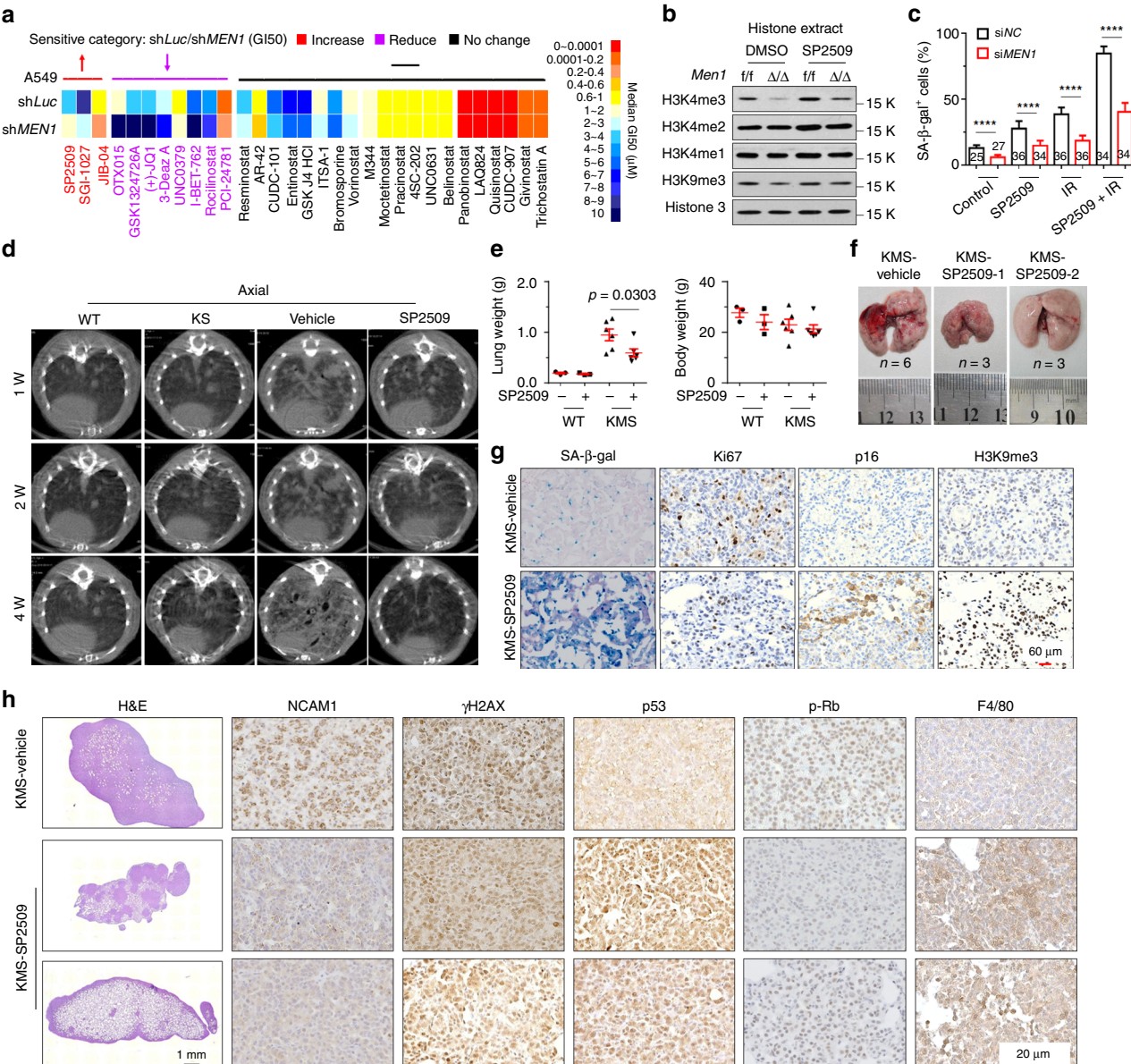

**Fig. 7 An LSD1 inhibitor reverses NE differentiation. a** Heatmap of median $GI_{50}$ values for A549-sh*Luc* and A549-sh*MEN1* cells treated with 96 small molecule inhibitor compounds for 3 days. The results were ranked according to the difference in sensitivity between A549-sh*Luc* and A549-sh*MEN1* cells as determined by the $GI_{50}$ ratios ($GI_{50}$ A549-sh*Luc*/A549-sh*MEN1*). **b** *Men1*^f/f and *Men1*^Δ/Δ MEF cells were treated with SP2509 for 24 h, and the levels of H3K4me3, H3K4me2, H3K4me1, and H3K9me3 modifications in histone extracts were determined via western blotting. **c** Quantification of the SA-β-gal-positive IMR-90 cells shown in Supplementary Fig. 7b. Cells were transfected with siRNA for 48 h and then treated with 1 μM SP2509 for 4 h. SA-β-gal staining was performed 2 days after exposure to 5 Gy of IR. **d** Axial microCT images in the indicated planes from WT, KS, KMS, and SP2509-treated (25 mg/kg i.p. injection) KMS mice at 1, 2, and 4 weeks. **e** Quantification of lung and body weight for KMS-vehicle and SP2509-treated KMS mice at 51 days (WT con: $n = 3$; WT SP2509: $n = 3$; KMS con: $n = 6$; KMS SP2509: $n = 6$). **f** Brightfield images of lung tissue from KMS and SP2509-treated KMS mice at 51 days. **g** Representative images of SA-β-gal staining and IHC staining in consecutive lung sections from KMS-vehicle, and SP2509-treated KMS mice at 51 days. Scale bars, 60 μm. **h** H&E images (left) and IHC staining for the indicated proteins in lung sections from KMS-vehicle and SP2509-treated KMS mice at 51 days ($n = 7$ per group). Scale bars, 1 mm (H&E) and 20 μm (IHC). Data are represented as mean ± SD in **c** and as mean ± SEM in **e**. Dots in **e** depict individual samples. Significance determined by Two-tailed unpaired $t$ tests in **c** and **e** are indicated. ****$p < 0.0001$. Results in **b** are representative of three independent experiments. Source Data are provided as a Source Data file.

**LSD1 inhibitor reverses NE differentiation in lung cancers.** Considering that chromatin histone remodeling is an essential step for NE differentiation, we further investigated whether epigenetic inhibitors could reverse the progression of NE differentiation. We performed a chemical screen using a commercial small molecular compound library of 96 compounds that target epigenetic regulators (Supplementary Table 2). Interestingly, A549-sh*MEN1* cells were most sensitive to histone demethylase

inhibitors, including SP2509 (an LSD1 inhibitor) and JIB-04 (a JMJD inhibitor), and SGI-1027 (a DNMT inhibitor), with half-lethal doses of 1.779, 2.495, and 0.38 μM, respectively (Fig. 7a). SP2509 specifically targets LSD1, which is responsible for inhibiting demethylation of H3K4me1 and H3K4me2 and promotes H3K4me3[31]. Cellular histone analysis showed that the deletion of *Men1* notably decreased H3K4me3, but not H3K4me1 and H3K4me2, and moderately diminished H3K9me3; however, these

effects were attenuated by treatment with SP2509 in MEF cells (Fig. 7b). 2-PCPA-1a, an inhibitor of LSD1, promotes oncogenic BRAF mutation-induced OIS in melanoma cells[29]. IR exposure elicited obvious SA-β-gal staining in IMR-90 cells (51.9%) that was eliminated by MI-3 (30.0%) but markedly increased by exposure to SP2509 (95.4%) (Supplementary Fig. 7a). The combination of SP2509 and MI-3 maintained similar levels of SA-β-gal staining in IMR-90 cells before and after IR exposure (Supplementary Fig. 7a). Reductions in SA-β-gal staining were also found in siMEN1-IMR-90 cells with or without exposure to IR, but staining was restored by treatment with SP2509 (Fig. 7c, Supplementary Fig. 7b). These results indicate that SP2509 effectively restores MEN1-related senescence.

Next, we sought to assess the pharmacological effect of SP2509 on spontaneous NE-type lung cancer in vivo (Supplementary Fig. 7c). Real-time microCT scanning indicated that KMS mice displayed visible dot-like markings and high lamellar density in lung regions at 1–4 weeks (Fig. 7d, Supplementary Fig. 7d). KMS mice treated with SP2509 showed normal lung markings similar to those of the asymptomatic or mildly symptomatic KS mice (Fig. 7d, Supplementary Fig. 7d). The mice were killed at day 51, and the lung weight, but not the body weight, of KMS mice was significantly elevated; this elevation was blocked by treatment with SP2509 (Fig. 7e). Morphological observations indicated that KMS mice developed overtly enlarged lungs with scattered tumor nodules (Fig. 7f). Strikingly, three of the six SP2509-treated mice exhibited tumor-free development, and small sporadic tumor nodules appeared on the lung surface in the other three SP2509-treated mice (Fig. 7f). As expected, SP2509 treatment notably restored the SA-β-gal staining, p16 and H3K9me3 levels, and reduced Ki67 expression in KMS mice (Fig. 7g, Supplementary Fig. 7e). The H&E results indicated that treatment with SP2509 notably reduced the tumor nodules in three mice, and peripheral hyperplasia or scarce tumor nodules were observed in the lungs of the other three mice (Fig. 7h). SP2509 treatment evidently reduced the expression of NCAM1 and moderately decreased the accumulation of γH2AX in lung tissues of KMS mice (Fig. 7h, Supplementary Fig. 7g). SP2509 also effectively attenuated the reduction in p53 and p-Rb by loss of menin in KMS mice (Fig. 7h, Supplementary Fig. 7g). Infiltration of macrophages was evidenced by increased staining of F4/80 in lung tissue from KMS mice treated with SP2509 (Fig. 7h, Supplementary Fig. 7g). Kaplan-Meier survival curves indicated that male KMS mice treated with SP2509 experienced long-term survival relative to KMS-vehicle mice ($p = 0.0127$, log-rank test) (Supplementary Fig. 7f). To delineate the role of SP2509 in lung cancer growth in vivo, A549-shLuc and A549-shMEN1 cells were subcutaneously transplanted into nude mice. Treatment of SP2509 significantly suppressed A549-shMEN1 tumor volume in vivo but did not obviously block A549-shLuc xenografts (Supplementary Fig. 7h). The reduction of menin significantly increased Ki67 index, decreased p16 and H3K9me3 staining in the A549-shMEN1-vehicle compared with A549-shLuc-vehicle xenograft tumors. Importantly, a significantly reduced Ki67 staining and increased p16 and H3K9me3 staining were observed in tumors from A549-shMEN1-SP2509 group, but not in tumors from wild-type A549-shLuc-SP2509 group (Supplementary Fig. 7i, j). These results demonstrate that SP2509 predominantly inhibit the growth of lung tumors with menin-low expression.

## Discussion

In the present GEMMs, spontaneous mixed-type lung cancers with typical NE characteristics occurred owing to Men1 single-gene deficiency. Specific depletion of Men1 in ATII cells markedly accelerated the initiation and progression of Kras[G12D]-driven lung cancer, notably shortening the survival time of mice to 9 weeks with more aggressive behavior. Importantly, the menin expression is evidently inactivated in certain human primary lung cancers, and this low expression is further associated with elevated NE profiles and significantly reduced 9-year overall survival. Combined with the previous reports[15,32,33], the present study further define a pivotal tumor suppressor function of MEN1 in lung tumorigenesis.

Poorly differentiated P-NETs arise either de novo or through the acquisition of genomic and epigenetic alterations from pre-existing NSCLCs as a mechanism of adaptive response to therapeutic stress[34]. Fundamental histological transformation from mesenchymal NSCLC to NE-type SCLC is observed in a subset of tyrosine kinase inhibitor-resistant NSCLCs[34]. In contrast, NE-type SCLC tumors initially treated successfully by chemotherapy can recur as chemoresistant variants and show progression toward a mesenchymal NSCLC phenotype[20]. These findings indicate that cross-talk between mesenchymal and NE cells strongly influences tumor cell behavior[20]. Although tumor heterogeneity is a recurrent feature of lung cancer, the key regulator underlying this histological differentiation remains undiscovered. Activating Hras-transduced cells upregulates the expression of vimentin and CD44 and silences the expression of NE markers, thereby resulting in an NE-to-mesenchymal transition in SCLC cells[20]. Our GEMMs revealed that the transition from an oncogenic Kras-induced mesenchymal phenotype to an NE phenotype could be induced by loss of Men1. The results strongly suggest that menin is required for EMT progression, which menin antagonizes the epithelial-to-NE transition, and that loss of menin could be a mechanism by which progenitor-like ATII cells differentiate toward a more NE cell-like state. Loss of menin may also be a mechanism for tumor cells to successfully escape from TIS, thus leading to tumor cell heterogeneity and resistance to treatment. We have elucidated the interplay between Ras and menin, which has an important role in controlling lung cancer malignancy[35]. Here, we further propose that menin and Kras cooperatively control the directional differentiation of epithelial cell-derived lung cancer cells.

An altered senescence program is one of the prominent features of NE tumors and is closely associated with aggressive clinical behavior in PanNETs[30,36]. Here, we demonstrated that the Kras activation-induced lung epithelial cell senescence program was almost completely blocked by the loss of menin during lung carcinogenesis. Thus, cells with inactivated menin pathways that carry stress-induced genomic damage can escape senescence programs and enter an uncontrolled one-way malignant transformation process. This process may be one of the mechanisms by which Men1 deletion dramatically accelerates lung tumorigenesis caused by Kras mutation. Specific inhibition of LSD1 has been found to effectively restore OIS and control the growth of melanoma that is resistant to BRAF inhibitors[29]. We found that the LSD1 inhibitor SP2509 effectively restored the chromatin H3K4me3 and H3K9me3 that was reduced by MEN1 deletion, inducing the senescence process and inhibiting NE differentiation. In the KMS GEMM, SP2509 effectively suppressed the development of lung cancer and upregulated p53/Rb pathways. We conclude that differentiation of epithelial cells requires the cells to pass the barrier of normal senescence program mediated by chromatin histone remodeling during lung carcinogenesis. Our discoveries strongly suggest that epigenetic targeting of LSD1 is an effective strategy for the treatment of menin-low NE-type lung cancer.

The present study defines an interesting epigenetic mechanism by which menin controls senescence and differentiation during lung carcinogenesis and provides a novel GEMM for further

investigating the pathogenesis of P-NETs in vivo (Supplementary Fig. 8).

## Methods

**Mouse models.** Animal welfare was ensured, and experimental procedures were performed in strict accordance with animal welfare and other related ethical regulations, and the procedures were approved by the Institutional Animal Care Committee of the Medical College at Xiamen University. The mice were housed in a temperature-controlled sterile room where humidity and light were carefully monitored and controlled. Intercrossed KMS (*LSL-Kras*$^{G12D/+}$; *Men1*$^{f/f}$; *Sftpc*-Cre) mice were maintained on a mixed background of B6.129 S4, B6.129 S (FVB), and B6.129S. Each group of mice had an equal number of males and females. At 6–8 weeks of age, all conditional *KO* mouse models received i.p. injections of 100 mg/kg TAM once a day for 5 days[37] followed by other treatments. The TAM was dissolved in corn oil containing 10% ethanol. The genotype was verified by PCR 1 week later. The genotyping primer sequences (listed in Supplementary Table 4) were synthesized by Sangon Biotech. For survival studies, as previous reported[21], the endpoints included but were not limited to labored and fast breathing, reduced eating or movement, energy loss, or weight loss > 20% of the initial body weight.

**Co-IP assays.** Cells were lysed in NP-40 lysis buffer supplemented with 1 µg/ml leupeptin, 1 µg/ml pepstatin, and 1 µg/ml phenylmethanesulfonyl fluoride. The lysates were then centrifuged, and 1 mg of protein lysate was loaded onto pre-washed protein A/G beads. The lysates were immunoprecipitated at 4 °C with the indicated antibodies. Next, the lysates were incubated with the protein A/G beads for 2 h at 4 °C. After centrifugation, the beads were washed three times with NP-40 buffer. The bound proteins were eluted with 2 × sodium dodecyl sulphate sample buffer and subjected to western blotting.

**ChIP assays.** ChIP assays were then carried out with the indicated antibodies according to the protocol of the ChIP Assay kit (DAM1442485; Millipore). In all, $1 \times 10^6$ MEFs or IMR-90 cells were treated with 1% formaldehyde for 10 min at 37 °C. Cell pellets were solubilized in ChIP lysis buffer, and followed by pulsed ultrasonication to shear cellular DNA and cleared by centrifugation at $12,000 \times g$ for 10 min. Part of the supernatant was digested with proteinase K (65 °C for 2 h); the DNA was isolated by spin columns; and input DNA was quantitated by real-time PCR. Equivalent amounts of chromatin were incubated with primary antibody overnight at 4 °C, and protein A-agarose beads were added and incubated 2 h at 4 °C. Immune complexes were washed in low- and high-salt ChIP buffers, eluted and incubated in NaCl (65 °C for 2 h), and then digested with proteinase K. DNA pulled down by the antibody was purified by phenol/chloroform extraction and purified DNA was quantitated by RT-qPCR using the Step One Plus real-time PCR system (Applied Biosystems). The antibodies and primer pairs (PP) sequences for ChIP assays are shown in Supplementary Table 3.

**Drug screening.** A549-sh*Luc* and A549-sh*MEN1* cell lines were cultured in Dulbecco's Modified Eagle Medium (DMEM) as described above. To screen a library of small molecular compounds targeting epigenetic regulators (96 compounds; Selleck, Supplementary Table 2), 2000 cells per well were seeded in 96-well plates for 24 h before treatment. The cells were treated with each compound at seven different concentrations (twofold serial dilution, typically from 10 µM to 156.25 nM) in duplicate wells for 3 days. A Cell Counting Kit 8 (CCK8) assay was used to evaluate cell viability, which was quantified using a Thermo Scientific Varioskan Flash instrument. The 50% growth inhibition (GI$_{50}$) values were calculated by using SPSS 17.0 software.

**In vivo treatment with compound SP2509.** *LSL-Kras*$^{G12D/+}$; *Men1*$^{f/f}$; *Sftpc*-Cre lung tumors were induced by TAM treatment. A week later, the mice received i.p. injections of 25 mg/kg SP2509 in 90% corn oil containing 10% dimethyl sulfoxide (DMSO)[31], vehicle as corn oil containing 10% DMSO. The KMS mice were treated with SP2509 twice a week for 3 weeks. Tumor initiation and progression were analyzed by microCT. The time points are indicated in Supplementary Fig. 7c.

**MicroCT.** Mice were scanned under isoflurane anesthesia using a small animal Quantum FX microCT scanner (Siemens Inveon PET/CT, Germany) at 1.4 mm resolution and 80 kV with a 500 µA current. Images were acquired with Inveon Acquisition Workplace 2.1 and IDL Virtual Machine 6.3. The scans were calibrated for Hounsfield Units (HU) by determining the mean values of "Bed" and "Air" for representative scans with the region of interest (ROI) tool and matching those values to their known HU (−800 HU and 1000 HU, respectively) using the Inveon Research Workplace IRW 4.2 tool. The examined location was lung tissue. Screenshots were obtained manually using ACDsee tools.

**Study of human primary lung cancer specimens.** This study was approved by the Medical Ethics Committee of Xiamen University and Jilin University, and written informed consent was obtained from all participants or from patients' representatives if direct consent could not be obtained. We collected 191 lung cancer specimens with tumor-adjacent lung tissues from patients who had undergone resection at the China-Japan Union Hospital of Jilin University and stored the tissues at the tissue bank of the China-Japan Union Hospital of Jilin University. The patients were diagnosed with adenocarcinoma, adenosquamous carcinoma, and SCLC by pathologists at the China-Japan Union Hospital, and diagnoses were made according to the World Health Organization criteria.

**Quantification and statistical analysis.** SPSS 17.0 was used to construct diagrams and perform statistical analyses. Drug GI$_{50}$ values were analyzed with two-tailed unpaired $t$ tests. Survival analyses were analyzed with log-rank (Mantel-Cox) test. IHC statistical analysis with Two-tailed unpaired $t$ tests. Clinicopathological parameters were analyzed using chi-square tests ($\chi^2$ tests). Each experiment was performed at least triplicate.

**Reporting summary.** Further information on research design is available in the Nature Research Reporting Summary linked to this article.

## Data availability

The source data underlying Figs. 1d, f, g; 2c, d; 3c–j; 4c, e; 5b–h; 6a, c–e; 7a–c, e; and Supplementary Figs. 1b; 2e, f; 3b–j; 4b, e; 5a, c–h; 6b–h; 7b, e–h, j, and Table 1 are provided as a Source Data file. All other original data that support the findings of this study are available from the corresponding author upon reasonable request.

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

## Acknowledgements

We thank the valuable comments from other members of our laboratories. We are grateful to Dr. Francis Collins for providing the heterozygous *Men1* locus (*Men1*$^{+/-}$) mice. We thank Dr. Patricia Ernst for providing the *MLL*$^{f/f}$ mice. This work was supported by grants from the National Natural Science Foundation of China 81572778 and U1605224 (to G.H.J.) and the Fundamental Research Funds for the Central Universities 20720150066 (to G.H.J.).

## Author contributions

Conceptualization, G.H.J.; methodology, G.H.J., H.Q., B.M.J.; formal analysis, G.H.J., H.Q., B.M.J.; investigation, G.H.J., H.Q., B.M.J.; B.X.; S.B.G.; L.Z.; L.Y.Z.; S.S.; J.B.Y.; X.L.; resources: Z.F.W.; visualization, G.H.J. H.Q, B.M.J.; writing–original draft, G.H.J., Q.F.Z., B.X., H.Q., B.M.J.; supervision and funding acquisition, G.H.J.

## Competing interests

The authors declare no competing interests.
