## [Peer Review File · Nature Communications]

Reviewers' comments:

Reviewer #1 (Remarks to the Author): Expert in oncogene induced senescence

The manuscript entitled "MEN1 deficiency leads to neuroendocrine differentiation of lung cancer by disrupting the DNA damage response and senescence progression" by Huan Qiu et al describes the involvement of MEN1 in restricting lung cancer by analysing the effect of combining oncogenic Kras expression with Men1 deficiency in animal models and cell lines. The main effects of Men1 deletion in the context of Kras-initiated lung cancer are the induction of a neuroendocrine (NE) phenotype and the disruption of the DNA damage and the senescence responses.

The manuscript provides a lot of information that sometimes is difficult to follow. It would be useful to have schematic diagrams explaining the main results. It is structured around three main phenotypes resulting from loss of Menin: accumulation of DNA damage, bypass of the senescence response and induction of a NE phenotype. The level of experimental evidence provided for each of these three phenotypes is variable. Altogether, their findings are interesting and might be relevant for the cancer field. However, the authors would need to improve their experiments and provide additional evidence.

Fig1 shows how loss of Men1 induces lung tumours with NE marker expression using heterozygous Men1^{+/-} mice and inducible knockout Men1 mice (Men1 Δ/Δ). Lungs show positive staining for NCAM1, NSE, SYN and CGA, and are also more proliferative.

Quantification of IHC stainings is done by labelling them as strongly positive, positive or negative. This is a very subjective quantification. Ideally tissue sections should be scanned and quantified using an appropriate histological analysis software.

Fig2 demonstrates the oncogenic cooperation between activation of oncogenic Kras and Men1 deficiency in lung carcinogenesis. Loss of Men1 results in accelerated tumour formation, decreased survival and a NE phenotype of the Kras-initiated tumours. Authors show, in agreement with previous reports, that Kras induction leads to the appearance of lesions composed of cells expressing mesenchymal markers (Nestin, Vimentin and ZEB1) and loss of epithelial ones (E-cadherin). In contrast, combined Men1 deletion reverses the mesenchymal markers, increasing E-cadherin and reducing Nestin, Vimentin and ZEB1, while at the same time induces the expression of NCAM1 and NSE, markers of NE phenotype. It would be desirable to show quantifications of all these stainings apart from showing representative pictures to get an idea of the magnitude of the changes.

Fig3 addresses the effect of Men1 loss on p53/Rb pathway. The authors claim that menin strongly upregulates the expression of p53 through an MDM2-mediated ubiquitination degradation pathway. This is based on the observed decreased expression of p53 and increased P-Rb (the inactive form of Rb) in tissues from mice deficient in Men1. However, the expression levels of p53 detected in wild type lungs are really strange since p53 is typically kept at very low levels under the control of Mdm2. In addition, the authors analysed the expression of these proteins by Western blot and checked the ubiquitylation and the stability of p53 after loss of Menin. Surprisingly, the authors do not show the levels of Mdm2 and refer to Mdm2 ubiquitylation on a supplementary figure. How are the levels of Mdm2 in Men1 deficient tissues, and in cells after knockdown or overexpression? Could the authors try to provide evidence of the post-translational effect on Rb that is simply referred as already published?

Then the authors try to identify the mechanism by which Menin alters p53 and Mdm2. Since Menin is not an E3 ubiquitin ligase, they check the levels of several E3 ligases after loss of Menin. One of these E3 ligases is bTrCP. Using gamma irradiation, they observe that bTrCP is increased in a Menin-dependent manner. The authors claim that bTrCP interaction with Mdm2 is lost in Menin deficient cells, although this reviewer cannot see such loss of interaction. Menin is shown to interact with regulatory regions in the bTrCP gene and supplementary figures (incorrectly labelled in the text) show that removal of Menin from bTrCP regulatory regions leads to altered pattern of H3K4 methylation. All these results lead the authors to propose that Menin epigenetically regulates the E3 ligase bTrCP to control the stability of MDM2/p53. However, the authors should alter the expression of bTrCP in the presence/absence of Menin to check the effect on p53 and Mdm2 to substantiate this claim.

Fig4 is an attempt to prove that the absence of Menin leads to DNA damage accumulation due to aberrant DNA damage repair. This would result from a role of Menin maintaining genomic instability through p53-dependent HR and p53-independent error-prone NHEJ mechanisms. In panel D, the authors show an IF for menin and gammaH2AX without quantifying the co-localization. This should be done. Panel E is used to try to show the altered expression of key DNA damage repair pathway proteins, but this reviewer cannot detect the claimed changes. Would it be possible to show Western blots with more clear changes and quantifications?

Fig5 deals with the role of menin in Kras-induced senescence. Senescence-associated beta-Galactosidase is used as a marker of senescence induction in vivo with no co-staining of other markers in the same slides or consecutive sections. This would be the ideal situation to judge the concomitant expression of senescence markers or the presence of senescence markers in cells showing no proliferation. Quantification of the SAbGal staining should be provided. Whole-mount SAbGal staining followed by paraffin embedding could help performing additional IHC markers and will also provide better quality histology. As additional IHC marker the authors show p16 expression. As judged by the pictures shown, p16 expression is huge in normal lung tissue a very strange finding. Could the authors show appropriate positive and negative controls for this IHC? Next, the authors decided to use MI-3 to inhibit Menin. Why did they not use their shMen1? In panel E, the authors show irradiated IMR90 cells, treated with MI-3 inhibitor and analysed for p16 mRNA levels 2 days after radiation. Has this huge increase in p16 mRNA after radiation ever been described? My recollection is that p16 might increase at very late time points after radiation but not after such a short period of time. In panel F a similar result is shown but this time for IL6, IL-1b and TGF-b. The only mRNA that is increased by radiation seems to be IL-1b, with IL-6 and TGF-b showing very modest changes (if any). So the decreased levels of these genes using MI-3 occur independently of the induction with radiation, they are also shown to happen relative to the vehicle treated cells. Panel G shows the growth curves of cells lacking Menin during serial passaging. Deletion of Menin is accomplished by tamoxifen treatment. Have the authors tried to add tamoxifen to cells entering senescence or even already growth arrested? Have the authors checked that they are actually lacking Menin in the cultures that progress and proliferate? In panel H, the authors show a SAbGal quantification of these same cell cultures. Could the author combine this staining with detection of Menin to try to show if those cells that are SAbGal positive in the Menin deleted cells are actually still expressing Menin? Same for panel I, with p16, TGF-b and IL-6. Fig6 shows the correlation between Menin deficiency and the poor prognosis and NE differentiation in human lung cancer. Panel A is just a sample of histological analyses performed on clinical samples without any substantial information. Quantifications for these stainings are shown on the other panels, but they were done manually and they are not presented as contingency tables with statistical analysis, so there is no way to draw a conclusion regarding the status of Menin and the expression of the cell identity markers.

Finally, Fig7 shows how the use of an LSD1 inhibitor can reverse NE phenotype in tumours lacking Menin. The authors performed a mini-screening with epigenetic drugs and focused their attention on SP2509, an LSD1 inhibitor that can revert the deficiency in H3K4me3 observed after loss of Menin. This is nicely shown in panel B. However, in panel C they try to address the effect of the inhibitor on senescence by analysing SAbGal on IMR90 cells after irradiation and they show how the inhibitor by itself seems to slightly induce senescence in cells but fails to do so to the same level in Menin knockdown cells. Panel G shows the effect of the inhibitor in vivo in restoring senescence induction in lung tumours. Again, this analysis would require additional markers, especially a proliferation marker to discriminate between proliferative and growth-arrested cells, since this is very important to understand the effect of the inhibitor. Furthermore, following these treated mice and checking tumour progression would be critical to judge the potential efficacy of the inhibitor for cancer treatment. Panel H analyses the expression of NE markers and tries to prove that the inhibitor reverts NE differentiation but judging from representative pictures is impossible to get an accurate idea of the effect of the inhibitor on NE differentiation.

General comment: there is a lack of statistical analysis for most of the figures.

Reviewer #2 (Remarks to the Author): Expert in DNA damage

the Nature Communication submission by Qiu et al., is an interesting and timely manuscript that contains important data for understanding the roles of MEN1 deficiency and how it leads to DNA damage, senescence and neuroendocrine differentiation of lung cancer. In general, the data shown are convincing and make great use of animal models with deficiencies in MEN1 in the context of K-ras alterations in lung cancer. The paper lacks how DNA damage is created, through increased ROS or other cell cycle alterations or deficiencies in DNA repair. In general the paper makes a number of leaps without really assessing a mechanistic bases for understanding the real roles of MEN1 in the phenomenon reported. Nevertheless, the study is well documented, makes good correlations of MEN1 loss in human disease with DNA damage, senescence and neuroendocrine lung cancer formation and should be reconsidered for publication after some revisions.

Introduction: The Introduction should be reconceived. It is very difficult to read that discussion and understand the overall purposes of the paper. There are important transitions made in the Introduction that are difficult to sift through and this reviewer was not quite sure what the overall goals of the paper were after reading the discussion.

Results: In general, the results section was well done. Experiments are well planned and have appropriate controls. The studies are based almost exclusively on knock out animal models with little real mechanistic data supporting many of the overall conclusions. Nevertheless, there are important data that are shown that should be reported. Below are some points that could be addressed that would fill in missing sections of the paper.

Figure 1

Panels C,D. These data should include standard error and statistical evaluations. Also, the data are difficult to read in this context.

Figure 3

Statements on decreased p53 function should be eliminated since the functional roles of p53 have not been delineated. Statements on decreased pRb function should also be eliminated since the appropriate tests for its function have not been elucidated.

Panel J. Statistical evaluations should ne included. P values should be included.

Effects of MEN1 loss on cell cycle checkpoint responses should be evaluated.

Figure 4

These data clearly show that there is increased DNA damage in cells deficient in MEN1, but the reason for this damage is not delineated.

-Data in Figure D should be quantified

-DNA repair rates should be included to find out if DNA repair is deficient. Rates of HR (Rad51) and NHEJ (gamma---H2AX) should included.

-The DNA damage used in this paper by 10 Gy IR is way to high and represents a supralethal dose of IR exposure. These doses are not acceptable and should be reduced to 1-3 Gy IR.

Effects of Men1 deficiency on ROS formation should be assessed as a possible function for the DNA damage seen

Effects of Men1 deficiency on cell cycle checkpoint responses should be assessed as a possible means for the DNA damage noted.

-Statements made about DNA repair should be eliminated.

Figure 5

-Cell cycle alterations in cells with or without Men1 should be included

-Dose of IR is ridiculous and is not worth publishing. 10 Gy is supra-lethal

-

Figure 6

Panel B-E, inclusion of standard error and p values are needed

Figure 7

-Panel C. Statistics should be included, where are p values?

-Panels G, H. Can these be quantified?

Conclusions: This paper reveals important data on the roles of Men1 in DNA damage, senescence and the formation of neuroendocrine lung cancer. Important correlations to human disease is shown. However, the mechanistic roles of Men 1 are not delineated and the reader is left to fairly large jumps in trying to understand connections between Men1 loss and how neuroendocrine lung cancers form. A few additional data on DNA repair capacity and cell cycle checkpoint responses, as well as inclusion of functional assessments of p53 and pRb are needed to fill in a few of the missing pieces of this paper.

Introduction: The Introduction should be reconceived. It is very difficult to read that discussion and understand the overall purposes of the paper. There are important transitions made in the Introduction that are difficult to sift through and this reviewer was not quite sure what the overall goals of the paper were after reading the discussion.

Results: In general, the results section was well done. Experiments are well planned and have appropriate controls. The studies are based almost exclusively on knock out animal models with little real mechanistic data supporting many of the overall conclusions.

Reviewer #3 (Remarks to the Author): Expert in mouse models of lung cancer

NCOMMS-19-07700

Qiu et al. "MEN1 deficiency leads to neuroendocrine differentiation of lung cancer by disrupting the DNA damage response and senescence progression "

The authors use a genetically engineered mouse model (GEMM) to study lung cancer in mice induced with a KRAS mutation and loss of function of MEN1. They find that combined mutations lead to increase in lung cancers, increase in tumors with neuroendocrine (NE) differentiation, decreased TP53 and Rb status in these NE tumors, increased accumulated DNA damage after radiation, and anti-tumor responses to a LSD1 inhibitor. They also examine human lung tumors for MEN1 expression level and find lower MEN1 is associated with impaired survival. Selected genetic functional studies are done in two human non-small cell lung cancer (NSCLC) lines and mouse embryonic fibroblasts (MEFs). They conclude: "Our discoveries strongly suggest that epigenetic targeting of LSD1 is an effective strategy for the treatment of menin low NE-type lung cancer. In addition, our findings suggest the potential application of LSD1 inhibitors for the treatment of NE-type neoplasms originating from multiple organs. The present study defines an interesting epigenetic mechanism by which menin controls senescence and differentiation during lung carcinogenesis and provides a novel GEMM for further investigating the pathogenesis of P-NETs in vivo. "

Comments to the Authors:

The studies are technically well done and clearly presented. However, with relevance to publication as a Nature Communication the central issues of this report are: 1. Do they provide substantial new information for a major role of MEN1 in NE lung cancers (even for a subset of these); 2. Do they provide a mechanistic understanding of such a role; 3. Have they identified new targeted therapy for lung cancers with loss of MEN1 function?

I believe they have provided some new mechanistic understanding for a potential MEN1 role - mainly the effect on TP53, RB and overcoming KRAS oncogene induced senescence, increase in DNA damage. However, they do not provide convincing data that MEN1 loss of function plays a major role in NE cancers of any type, and they do not have data that using LSD1 inhibitors have major importance in preclinical model responses for MEN1 loss of function human NE tumors.

The major issue is what is the frequency of MEN1 loss in lung cancers, NE lung cancers, and KRAS mutant lung cancers? The available deposited data indicate these subsets must be very uncommon. While the authors should have presented this information, one can go into important databases and discover: there are no MEN1 mutations in the ~90 small cell lung cancers (SCLCs) in the George data set; 6 MEN1 mutations are present in lung adenocarcinomas (>600) in the TCGA dataset - 3 of which have a KRAS mutation; 3 MEN1 mutations in squamous lung cancers (>500) none with KRAS mutations; and searching deposited mutation datasets on human lung cancer lines there are very few MEN1 mutations found - but such SCLC and NSCLC preclinical models that can be tested do exist. Particularly since the authors focused on a KRAS based GEMM for their studies, the combination of KRAS and MEN1 inactivation in NSCLC are essentially rare. In any event, for the therapeutic aspects with an LSD1 inhibitor we need to see effects of such treatments on human preclinical models with patient acquired KRAS mutations and loss of MEN1 function both in vitro and in vivo (xenografts).

POINT-BY-POINT RESPONSE TO THE REVIEWERS 1' COMMENTS:

Q1: The manuscript entitled “*MEN1* deficiency leads to neuroendocrine differentiation of lung cancer by disrupting the DNA damage response and senescence progression” by Huan Qiu et al describes the involvement of *MEN1* in restricting lung cancer by analysing the effect of combining oncogenic Kras expression with *Men1* deficiency in animal models and cell lines. The main effects of *Men1* deletion in the context of Kras-initiated lung cancer are the induction of a neuroendocrine (NE) phenotype and the disruption of the DNA damage and the senescence responses.

The manuscript provides a lot of information that sometimes is difficult to follow. It would be useful to have schematic diagrams explaining the main results. It is structured around three main phenotypes resulting from loss of menin: accumulation of DNA damage, bypass of the senescence response and induction of a NE phenotype. The level of experimental evidence provided for each of these three phenotypes is variable. Altogether, their findings are interesting and might be relevant for the cancer field. However, the authors would need to improve their experiments and provide additional evidence.

Response: We appreciate reviewer’s encouraging comment that the study is interesting. According to the reviewer’s proposed experiments and suggestions, we carefully revised the entire manuscript, and attached the schematic diagram in the supplemental Fig.8. Hopefully this schematic diagram will help reviewers and readers understand the main idea of the text.

Q2: Fig1 shows how loss of *Men1* induces lung tumours with NE marker expression using heterozygous *Men1*^{+/-} mice and inducible knockout *Men1* mice (*Men1*^{ΔΔ}). Lungs show positive staining for NCAM1, NSE, Syn and CgA, and are also more proliferative. Quantification of IHC staining is done by labelling them as strongly positive, positive or negative. This is a very subjective quantification. Ideally tissue sections should be scanned and quantified using an appropriate histological analysis software.

Response: We apologize for the error and appreciate the insightful suggestions from the reviewer. According to the reviewer’s suggestions, the tissue sections were scanned and quantified using an appropriate histological analysis software. We have made according changes in Fig. 1d,f,g and Supplementary Fig. 1b as suggested by the

reviewer.

For automatic quantification, digital images of stained tissues were captured with a Motic microscope using DigiLabII-Client software. Quantification of IHC staining was calculated by image pro plus software. The same index for the same batch of samples uses the unified parameters. All the indexes were measured by positive area, the parameter as (H: 0-30, S: 0-180, I: 0-180 or H: 0-20, S: 0-160, I: 0-160). For automatic quantification the area of positive cells per sample, we randomly analyzed three fields (400 times magnification, 800x600 pixel) per individual sample, then calculated the mean value for each sample. The mean value and stand error of mean were used to calculate for each group. At last, the p-value was calculated by Two-tailed unpaired t tests.

Q3: Fig 2 demonstrates the oncogenic cooperation between activation of oncogenic *Kras* and *Men1* deficiency in lung carcinogenesis. Loss of *Men1* results in accelerated tumour formation, decreased survival and a NE phenotype of the *Kras*-initiated tumours. Authors show, in agreement with previous reports, that *Kras* induction leads to the appearance of lesions composed of cells expressing mesenchymal markers (Nestin, Vimentin and ZEB1) and loss of epithelial ones (E-cadherin). In contrast, combined *Men1* deletion reverses the mesenchymal markers, increasing E-cadherin and reducing Nestin, Vimentin and ZEB1, while at the same time induces the expression of NCAM1 and NSE, markers of NE phenotype. It would be desirable to show quantifications of all these staining apart from showing representative pictures to get an idea of the magnitude of the changes.

Response: According to reviewer's suggestion, we performed IHC quantification and statistical analysis as shown in Supplementary Fig. 2e,f. The IHC quantification and statistical analysis indicated that high staining Ki67 proliferation indexes and NE markers (NCAM1, NSE, CgA, Syn), in KMS lung tumors. The tumors in KS mice exhibited low staining for the epithelial marker E-cadherin and showed significantly high levels of mesenchymal markers (Nestin, Vimentin and ZEB1), whereas *Men1* deficiency in KMS mice clearly recovered E-cadherin staining and significantly reduced the expression of Nestin, Vimentin and ZEB1 in lung tumors (Supplementary Fig. 2e). The IHC quantification and statistical analysis indicated that staining of F4/80 was significantly blocked by *Men1* deficiency. The expression of CD31 and HMGA2 was evidently increased in KMS lung tumors (Supplementary Fig. 2f).

Q4: Fig 3 addresses the effect of *Men1* loss on p53/Rb pathway. The authors claim that menin strongly upregulates the expression of p53 through an

MDM2-mediated ubiquitination degradation pathway. This is based on the observed decreased expression of p53 and increased p-Rb (the inactive form of Rb) in tissues from mice deficient in *Men1*. However, the expression levels of p53 detected in wild type lungs are really strange since p53 is typically kept at very low levels under the control of MDM2. In addition, the authors analysed the expression of these proteins by Western blot and checked the ubiquitylation and the stability of p53 after loss of menin. Surprisingly, the authors do not show the levels of MDM2 and refer to MDM2 ubiquitylation on a supplementary figure. How are the levels of MDM2 in *Men1* deficient tissues, and in cells after knockdown or overexpression? Could the authors try to provide evidence of the post-translational effect on Rb that is simply referred as already published? Then the authors try to identify the mechanism by which menin alters p53 and Mdm2. Since menin is not an E3 ubiquitin ligase, they check the levels of several E3 ligases after loss of menin. One of these E3 ligases is bTrCP. Using gamma irradiation, they observe that bTrCP is increased in a menin -dependent manner. The authors claim that bTrCP interaction with MDM2 is lost in menin deficient cells, although this reviewer cannot see such loss of interaction. menin is shown to interact with regulatory regions in the bTrCP gene and supplementary figures (incorrectly labelled in the text) show that removal of menin from bTrCP regulatory regions leads to altered pattern of H3K4 methylation. All these results lead the authors to propose that menin epigenetically regulates the E3 ligase bTrCP to control the stability of MDM2/p53. However, the authors should alter the expression of bTrCP in the presence/absence of menin to check the effect on p53 and MDM2 to substantiate this claim.

Response: Q4 can be further divided into 5 sub-questions, and our responses were detailed as followings.

Q4-1: Fig 3 addresses the effect of *Men1* loss on p53/Rb pathway. The authors claim that menin strongly upregulates the expression of p53 through an MDM2-mediated ubiquitination degradation pathway. This is based on the observed decreased expression of p53 and increased p-Rb (the inactive form of Rb) in tissues from mice deficient in *Men1*. However, the expression levels of p53 detected in wild type lungs are really strange since p53 is typically kept at very low levels under the control of MDM2.

Response: We appreciate the insightful suggestions from the reviewer. We apologize for

this error. According to reviewer's suggestion, we have repeated p53 IHC staining by shortening the time of DAB staining. IHC results show that p53 expression was clearly decreased in *Men1*^{+/-} lung tumor. The correction has been made in Fig.3a.

Q4-2: In addition, the authors analysed the expression of these proteins by Western blot and checked the ubiquitylation and the stability of p53 after loss of menin. Surprisingly, the authors do not show the levels of MDM2 and refer to MDM2 ubiquitylation on a supplementary figure. How are the levels of MDM2 in Men1 deficient tissues, and in cells after knockdown or overexpression?

Response: According to reviewer's suggestion, the IHC and Western blot detection of MDM2 was performed, and the results were shown in Supplementary Fig. 3a,b. The levels of MDM2 protein in lung tissues of *Men1*^{+/-} (Supplementary Fig. 3a) and *Men1*^{Δ/Δ} mice (data not shown in manuscript, below) were obviously increased. Furthermore, the Western blot analysis indicated that the expression of MDM2 protein was strongly suppressed by overexpression of *MEN1* in A549 cells (Supplementary Fig. 3b).

Q4-3: Could the authors try to provide evidence of the post-translational effect on Rb that is simply referred as already published?

Response: We appreciate this comment from the reviewer. It has been reported that *Men1* regulates Rb expression in a posttranscriptional manner (1). Consistently, we observed that *KD* of *MEN1* reduced the expression of Rb protein and upregulate the phosphorylation of Rb (p-Rb), without altering the mRNA levels of *Rb* in A549 cells (Supplementary Fig. 3i). Interestingly, the half-life of p-Rb, but not Rb, was prolonged from <1 to about 6 hours by *KD* of *MEN1* in A549 cells (Supplementary Fig. 3j). These results indicated that *MEN1* regulated Rb activity by controlling its posttranslational modification. The correction has been made in Results section (page 9), and now the data is presented in Supplementary Fig. 3i,j.

Q4-4: Then the authors try to identify the mechanism by which menin alters p53 and Mdm2. Since menin is not an E3 ubiquitin ligase, they check the levels of several E3 ligases after loss of menin. One of these E3 ligases is bTrCP. Using gamma irradiation, they observe that bTrCP is increased in a menin -dependent manner. The authors claim

that bTrCP interaction with MDM2 is lost in menin deficient cells, although this reviewer cannot see such loss of interaction.

Response: We appreciate this comment from the reviewer and apologize that the description of our results was not sufficiently clear. We have carefully revised the manuscript with special attention to improve the expressions: “The interaction between MDM2 and β TrCP, but not that of MDM2 and MDMX, was **diminished** by deletion of *Men1* in MEF cells” (Page 9).

Q4-5: menin is shown to interact with regulatory regions in the bTrCP gene and supplementary figures (incorrectly labelled in the text) show that removal of menin from bTrCP regulatory regions leads to altered pattern of H3K4 methylation. All these results lead the authors to propose that menin epigenetically regulates the E3 ligase bTrCP to control the stability of MDM2/p53. However, the authors should alter the expression of bTrCP in the presence/absence of menin to check the effect on p53 and MDM2 to substantiate this claim.

Response: According to the reviewer’s suggestion, we further investigated if the β TrCP is necessary for menin-regulated MDM2/p53 pathways. We transfected A549-vector or A549-*MEN1* cells with si β TrCP for 48 hours. The Western blot result clearly showed that upregulation of β TrCP and p53, and reduction of MDM2 by menin overexpression were counteracted by siRNA-mediated *KD* of β TrCP in A549 cells (Supplementary Fig. 3h). This result suggests that menin regulates MDM2/p53 pathways at least partly through β TrCP.

Q5: Fig 4 is an attempt to prove that the absence of menin leads to DNA damage accumulation due to aberrant DNA damage repair. This would result from a role of menin maintaining genomic instability through p53-dependent HR and p53-independent error-prone NHEJ mechanisms. In panel D, the authors show an IF for menin and gammaH2AX without quantifying the co-localization. This should be done. Panel E is used to try to show the altered expression of key DNA damage repair pathway proteins, but this reviewer cannot detect the claimed changes. Would it be possible to show Western blots with more clear changes and quantifications?

Response: According to the reviewer’s suggestion, we quantified the co-localization of menin and γ H2AX. The GraphPad Prism 5 software was used to determine statistical analysis by Mann–Whitney U test. Now the data is presented in Supplementary Fig. 5a.

We observed that both sh*MEN1* and *Men1* ^{$\Delta\Delta$} obviously enhanced the expression of

DDR factors such as RAD51, Ku70/80 and DNA-PKcs proteins in A549 and MEF cells, respectively. These data suggest that loss of *Men1* leads to aberrant DNA damage response, further resulting in accumulation of DNA damage. According to the reviewer's suggestion, we repeated the analysis of DNA damage repair pathway proteins using independently generated cell lines. The A549-sh*Luc* and A549-sh*MEN1* cells, *Men1*^{fl/fl} and *Men1*^{Δ/Δ} MEF cells were radiated with 10 Gy of IR and collected 1 hour after IR. Western blot was repeatedly conducted to detect the expression of indicated protein and the correction has been made in Fig.4e.

Q6: Fig 5 deals with the role of menin in Kras-induced senescence. Senescence-associated beta-Galactosidase is used as a marker of senescence induction in vivo with no co-staining of other markers in the same slides or consecutive sections. Senescence markers or the presence of senescence markers in cells showing no proliferation. Quantification of the SAbGal staining should be provided. Whole-mount SAbGal staining followed by paraffin embedding could help performing additional IHC markers and will also provide better quality histology. As additional IHC marker the authors show p16 expression. As judged by the pictures shown, p16 expression is huge in normal lung tissue a very strange finding. Could the authors show appropriate positive and negative controls for this IHC? Next, the authors decided to use MI-3 to inhibit menin. Why did they not use their sh*Men1*? In panel E, the authors show irradiated IMR90 cells, treated with MI-3 inhibitor and analysed for p16 mRNA levels 2 days after radiation. Has this huge increase in p16 mRNA after radiation ever described? My recollection is that p16 might increase at very late time points after radiation but not after such a short period of time. In panel F a similar result is shown but this time for IL6, IL-1b and TGF-β. The only mRNA that is increased by radiation seems to be IL-1b, with IL-6 and TGF-β showing very modest changes (if any). So the decreased levels of these genes using MI-3 occur independently of the induction with radiation, they are also shown to happen relative to the vehicle treated cells. Panel G shows the growth curves of cells lacking menin during serial passaging. Deletion of menin is accomplished by tamoxifen treatment. Have the authors tried to add tamoxifen to cells entering senescence or even already growth arrested? Have the authors checked that they are actually lacking menin in the cultures that progress and proliferate? In panel H, the

authors show a SAbGal quantification of these same cell cultures. Could the author combine this staining with detection of menin to try to show if those cells that are SAbGal positive in the menin deleted cells are actually still expressing menin? Same for panel I, with p16, TGF- β and IL-6.

Response: Q6 can be further divided into 6 sub-questions, and our responses were detailed as followings.

Q6-1: Fig 5 deals with the role of menin in Kras-induced senescence. Senescence-associated beta-Galactosidase is used as a marker of senescence induction in vivo with no co-staining of other markers in the same slides or consecutive sections. Senescence markers or the presence of senescence markers in cells showing no proliferation. Quantification of the SAbGal staining should be provided. Whole-mount SAbGal staining followed by paraffin embedding could help performing additional IHC markers and will also provide better quality histology.

Response: According to the reviewer's suggestion, we performed IHC staining with consecutive frozen section. We observed that deletion of *Men1* reduced SA- β -gal staining and p16 expression, and increased Ki67 expression in lung tissues of KMS mice compared with KS mice. We appreciate this comment from the reviewer, and now the data is presented in Fig.5a,b.

Q6-2: As additional IHC marker the authors show p16 expression. As judged by the pictures shown, p16 expression is huge in normal lung tissue a very strange finding. Could the authors show appropriate positive and negative controls for this IHC?

Response: We appreciate reviewer's comments, the IHC detection of p16 was repeated with independent tissue samples. In order to confirm whether the low expression of p16 in *Men1* ^{$\Delta\Delta$} group was accurate, the staining time was extended to exclude the false negative of *Men1* ^{$\Delta\Delta$} group. The IHC results indicated that expression of p16 was significantly repressed by the absence of *Men1* in *Men1* ^{$\Delta\Delta$} mice. The correction has been made in Fig.5c.

Q6-3: Next, the authors decided to use MI-3 to inhibit menin. Why did they not use their sh*Men1*?

Response: Real time qPCR results showed that siRNA-mediated KD of *MEN1* clearly reduced SA- β -gal staining in IMR-90 cells with or without IR exposure (Supplementary Fig. 7b).

Q6-4: In panel E, the authors show irradiated IMR90 cells, treated with MI-3 inhibitor and analysed for p16 mRNA levels 2 days after radiation. Has this huge increase in p16

mRNA after radiation ever described? My recollection is that p16 might increase at very late time points after radiation but not after such a short period of time. In panel F a similar result is shown but this time for IL6, IL-1b and TGF-b. The only mRNA that is increased by radiation seems to be IL-1b, with IL-6 and TGF-b showing very modest changes (if any). So the decreased levels of these genes using MI-3 occur independently of the induction with radiation, they are also shown to happen relative to the vehicle treated cells.

Response: We fully agree with the reviewer's concerns and explored this aspect according to reviewer's suggestion. To address reviewer's question, we treated the IMR-90 cells with 10 μ M MI-3 for 48 h, which was followed by cell collection on day 2, day 3, and day 4 after exposure to 3 Gy of IR. RT-qPCR showed that the mRNA levels of p16 were obviously increased on day 2 after IR, while the mRNA levels of IL-6, IL-1 β and TGF- β were increased at days 3 after IR, and the effects were able to be further neutralized by MI-3 treatment. The statistical analysis showed that the mRNA expression of p16 was significantly stimulated from day 2 after exposure of 3Gy IR, and notably blocked by the presence of MI-3 in IMR-90 cells ($p < 0.05$) (data not shown in manuscript, below). Our data similar to previous reports that the expression of p16 protein obviously up-regulated in primary human embryonic lung fibroblasts (MRC5) on day 1 and day 2 after exposure to 2.5 Gy or 10 Gy IR (2). Moreover, ROS-mediated DNA damage significantly promoted the expression of p16 and secretion of IL-6 in L6 rat myoblasts on day 2 and day 3 after H₂O₂ treatment, respectively (3).

Q6-5: Panel G shows the growth curves of cells lacking menin during serial passaging. Deletion of menin is accomplished by tamoxifen treatment. Have the authors tried to add tamoxifen to cells entering senescence or even already growth arrested? Have the authors checked that they are actually lacking menin in the cultures that progress and proliferate?

Response: We appreciate the insightful suggestions from the reviewer. During revision, we analyzed the potential effect of tamoxifen on induction of primary MEF cell senescence. Primary P6 *Men1^{fl/fl}* MEF cells were treated with TAM for 3 days and SA- β -gal stain results indicated that TAM treatment was not able to cause statistical difference in the number of

SA- β -gal⁺ cells. This is consistent with previous report that the senescence and proliferation of MEF cells were not affected by 4-hydroxytamoxifen, a metabolite of tamoxifen (4).

Q6-6: In panel H, the authors show a SAbGal quantification of these same cell cultures. Could the author combine this staining with detection of menin to try to show if those cells that are SAbGal positive in the menin deleted cells are actually still expressing menin? Same for panel I, with p16, TGF- β and IL-6.

Response: Next, we further investigated the effects of menin deletion on senescence in primary MEF cells. The *Men1^{fl/fl}* and *Men1 $\Delta\Delta$* primary MEF cells at passage 1, 6, and 9 were collected and Western blot results showed that the menin protein was undetectable in *Men1 $\Delta\Delta$* MEF cells; the expression of p16 gradually increased and reduced by menin deletion during culture progression. These results indicated that tamoxifen induced permanently *Men1* knockout in MEF cells as shown in below.

Q7: Fig 6 shows the correlation between menin deficiency and the poor prognosis and NE differentiation in human lung cancer. Panel A is just a sample of histological analyses performed on clinical samples without any substantial information. Quantifications for these staining are shown on the other panels, but they were done manually and they are not presented as contingency tables with statistical analysis, so there is no way to draw a conclusion regarding the status of menin and the expression of the cell identity markers.

Response: We fully agree with the reviewer's comment. According to reviewer's suggestion, the IHC quantification and statistical analysis were supplemented. The IHC results clearly showed that menin was stained in the nucleus and that menin expression in tumors, compared to that in adjacent lung tissues, was markedly reduced or undetectable in 42 out of the 157 NSCLC cases (26.8%) and in 15 out of the 34 SCLC cases (44.1%). Compared with those from NSCLC patients, samples from SCLC patients possessed significantly strong NE profiles and exhibited reduced mesenchymal characteristics (Fig. 6b). Furthermore, the menin-low (-) patients had significantly high staining for NE markers (NCAM1, NSE) but low staining for Vimentin compared with menin-high (+) patients (Fig.

6c-d). The correction has been made in Fig.6b,c,d.

Q8: Finally, Fig7 shows how the use of an LSD1 inhibitor can reverse NE phenotype in tumours lacking menin. The authors performed a mini-screening with epigenetic drugs and focused their attention of SP2509, an LSD1 inhibitor that can revert the deficiency in H3K4me3 observed after loss of menin. This is nicely shown in panel B. However, in panel C they try to address the effect of the inhibitor on senescence by analysing SAbGal on IMR90 cells after irradiation and they show how the inhibitor by itself seems to slightly induce senescence in cells but fails to do so to the same level in menin knockdown cells. Panel G shows the effect of the inhibitor in vivo in restoring senescence induction in lung tumors. Again, this analysis would require additional markers, especially a proliferation marker to discriminate between proliferative and growth-arrested cells, since this is very important to understand the effect of the inhibitor. Furthermore, following these treated mice and checking tumor progression would be critical to judge the potential efficacy of the inhibitor for cancer treatment. Panel H analyses the expression of NE markers and tries to prove that the inhibitor reverts NE differentiation but judging from representative pictures is impossible to get an accurate idea of the effect of the inhibitor on NE differentiation.

Response: Q8 can be further divided into 3 sub-questions, and our responses were detailed as followings.

Q8-1: Finally, Fig7 shows how the use of an LSD1 inhibitor can reverse NE phenotype in tumours lacking menin. The authors performed a mini-screening with epigenetic drugs and focused their attention of SP2509, an LSD1 inhibitor that can revert the deficiency in H3K4me3 observed after loss of menin. This is nicely shown in panel B. However, in panel C they try to address the effect of the inhibitor on senescence by analysing SAbGal on IMR90 cells after irradiation and they show how the inhibitor by itself seems to slightly induce senescence in cells but fails to do so to the same level in menin knockdown cells. Panel G shows the effect of the inhibitor in vivo in restoring senescence induction in lung tumors. Again, this analysis would require additional markers, especially a proliferation marker to discriminate between proliferative and growth-arrested cells, since this is very important to understand the effect of the inhibitor.

Response: According to the reviewer's suggestion, we performed IHC staining with consecutive frozen section. We observed that SP2509 treatment significantly increased

SA- β -gal staining and decreased Ki67 expression in lung tissues of KMS mice. We appreciate this comment from the reviewer, and now the data is presented in Fig.7g and Supplementary Fig. 7f.

Q8-2: Furthermore, following these treated mice and checking tumor progression would be critical to judge the potential efficacy of the inhibitor for cancer treatment.

Response: We thank the reviewer for pointing out this aspect. The Kaplan-Meier survival curves show that treatment of SP2509 was significantly prolonged the median survival from 42 days to 74 days in KMS mice. Now the data is presented in Supplementary Fig. 7e.

Q8-3: Panel H analyses the expression of NE markers and tries to prove that the inhibitor reverts NE differentiation but judging from representative pictures is impossible to get an accurate idea of the effect of the inhibitor on NE differentiation.

Response: We appreciate the insightful suggestions from the reviewer and have performed the IHC quantification and statistical analysis. The results show that SP2509 treatment significantly reduced the expression of NCAM1 and decreased the accumulation of γ H2AX in lung tissues of KMS mice. SP2509 also effectively attenuated the reduction in p53 and the inactivation of Rb (p-Rb) in KMS mice. Infiltration of macrophages was evidenced by increased staining of F4/80 in lung tissue from KMS mice treated with SP2509. The statistical significances were indicated in Supplementary Fig. 7g.

General comment: there is a lack of statistical analysis for most of the figures.

Response: We fully agree with the reviewer's comment. We have performed statistical analysis and corrected this issue in all figures.

References

1. Ivo, D., Corset, L., Desbourdes, L., Gaudray, P. & Weber, G. Menin controls the concentration of retinoblastoma protein. *Cell Cycle* **10**, 166-168 (2011).
2. Gabriel, K *et al.* To senesce or not to senesce: how primary human fibroblasts decide their cell fate after DNA damage. *Aging (Albany NY)* **8**, 158-176 (2016).
3. Gireedhar, V., Uttam, S. & Marie-Véronique, C. Replication stress-induced endogenous DNA damage drives cellular senescence induced by a sub-lethal oxidative stress. *Nucleic Acids Res* **45**, 10564-10582 (2017).
4. Mohamad, H *et al.* p53 Restoration in induction and maintenance of senescence: differential effects in premalignant and malignant tumor cells. *Mol Cell Biol* **36**, 438-451 (2016).

POINT-BY-POINT RESPONSE TO THE REVIEWERS 2' COMMENTS:

The Nature Communication submission by Qiu et al., is an interesting and timely manuscript that contains important data for understanding the roles of *MEN1* deficiency and how it leads to DNA damage, senescence and neuroendocrine differentiation of lung cancer. In general, the data shown are convincing and make great use of animal models with deficiencies in *MEN1* in the context of *K-ras* alterations in lung cancer. The paper lacks how DNA damage is created, through increased ROS or other cell cycle alterations or deficiencies in DNA repair. In general the paper makes a number of leaps without really assessing a mechanistic bases for understanding the real roles of *MEN1* in the phenomenon reported. Nevertheless, the study is well documented, makes good correlations of *MEN1* loss in human disease with DNA damage, senescence and neuroendocrine lung cancer formation and should be reconsidered for publication after some revisions.

Q1: The Introduction should be reconceived. It is very difficult to read that discussion and understand the overall purposes of the paper. There are important transitions made in the Introduction that are difficult to sift through and this reviewer was not quite sure what the overall goals of the paper were after reading the discussion.

Response: We appreciate reviewer's comments that the study is interesting yet the writing of manuscript needs to be improved. We have carefully revised the entire manuscript with special attention to improve the expressions, and the correction has been highlighted in the manuscript.

Q2: Figure 1 Panels C, D. These data should include standard error and statistical evaluations. Also, the data are difficult to read in this context.

Response: We apologize for the error and we appreciate the insightful suggestions from the reviewer. According to the reviewer's suggestions, the tissue sections were scanned and quantified using an appropriate histological analysis software. We have made according changes in Fig. 1d,f,g and Supplementary Fig. 1b as suggested by the reviewer.

For automatic quantification, digital images of stained tissues were captured with a Motic microscope using DigiLabII-Client software. Quantification of IHC staining was calculated by image pro plus software. The same index for the same batch of samples

uses the unified parameters. All the indexes were measured by positive area, the parameter as (H: 0-30, S: 0-180, I: 0-180 or H: 0-20, S: 0-160, I: 0-160). For automatic quantification the area of positive cells per sample, we randomly analyzed three fields (400 times magnification, 800x600 pixel) per individual sample, then calculated the mean value for each sample. The mean value and stand error of mean were used to calculate for each group. At last, the p value was calculated by Two-tailed unpaired t test.

Q3: Figure 3 statements on decreased p53 function should be eliminated since the functional roles of p53 have not been delineated. Statements on decreased p-Rb function should also be eliminated since the appropriate tests for its function have not been elucidated.

Response: We thank the reviewer for pointing out this aspect. According to reviewer's suggestion, we have carefully revised the manuscript with special attention to improve the expressions. "These findings suggest that menin-mediated control of NE differentiation is dependent on the p53/Rb pathway but not the MYC pathway" was deleted from our paper (page 8).

Q4: Panel J. Statistical evaluations should be included. p values should be included.

Response: According to the reviewer's suggestion, we have performed Two-tailed unpaired t test statistical analysis with the SPSS statistical software. The statistical results were indicated in the Figure 3j and Supplementary Fig. 3g.

Q5: Figure 4 These data clearly show that there is increased DNA damage in cells deficient in *MEN1*, but the reason for this damage is not delineated.

Response: We appreciate the insightful suggestions from the reviewer. To address reviewer's question, we further investigated how menin influences DNA damage response. We observed that DNA repair rates (RAD51/ γ H2AX) decreased in *Men1* ^{Δ/Δ} relative to *Men1*^{fl/fl} MEF cells (Supplementary Fig. 5e). Loss of *Men1* did not impact on the generation of reactive oxygen species (ROS), an inducer of DNA damage, in MEF cells after IR (Supplementary Fig. 5f). Deletion of *Men1* distinctly activated phosphorylation of cell cycle checkpoints p-CHK1 and p-CHK2 induced by IR exposure in MEF cells (Supplementary Fig. 5g). Our data suggest that menin maintains genomic stability at least partly by controlling cell cycle checkpoint, but not by generating endogenous ROS. The correction has been made in Results section (page 11), and now the data is presented in Supplementary Fig. 5e,f.

Q7: Data in Figure 4D should be quantified

Response: According to the reviewer's suggestion, we quantified the co-localization of menin and γ H2AX. The GraphPad Prism 5 software was used to determine statistical analysis by Mann–Whitney U test. The statistical results were indicated in the Supplementary Fig. 5a.

Q8: DNA repair rates should be included to find out if DNA repair is deficient. Rates of HR (Rad51) and NHEJ (γ H2AX) should be included.

Response: We thank the reviewer for pointing out this aspect. To address reviewer's concerns, *Men1^{fl/fl}* and *Men1 $\Delta\Delta$* MEF cells were exposed to 3 Gy of IR and collected at the indicated time points after IR exposure. Western blot analysis indicated that the expression of γ H2AX and RAD51 proteins was stimulated by IR. The DNA repair rates (RAD51/ γ H2AX) were decreased in *Men1 $\Delta\Delta$* relative to *Men1^{fl/fl}* MEF cells (Supplementary Fig. 5e). Our data suggest that deletion of *Men1* induces DNA damage accumulation at least partly via disordering DNA damage response. Now the data is presented in Supplementary Fig. 5e.

Q9: The DNA damage used in this paper by 10 Gy IR is way too high and represents a supralethal dose of IR exposure. These doses are not acceptable and should be reduced to 1-3 Gy IR.

Response: We fully agree with the reviewer's comment that 10 Gy IR is way too high and represents a supralethal dose of IR exposure. According to the reviewer's suggestion, *Men1^{fl/fl}* and *Men1 $\Delta\Delta$* mice were irradiated with modest dose (3 Gy) and sacrificed at one hour after exposure of IR. IF results shown that γ H2AX was obviously activated, and further aggravated by loss of *Men1*. Similar results were also found in *Men1*-deleted MEF cells at indicated time points after 3 Gy of IR. Now the data is presented in Supplementary Fig. 4a,b,c.

Q10: Effects of *Men1* deficiency on ROS formation should be assessed as a possible function for the DNA damage seen.

Response: We would like to thank the reviewer for their precious advice. The *Men1^{fl/fl}* and *Men1 $\Delta\Delta$* MEF cells were treated with 3 Gy IR. The MEF cells were collected and stained with DCFDA, and flow cytometry analysis was performed to assess the generation of ROS. We observed that production of ROS was not affected by *Men1* deletion as shown in Supplementary Fig. 5f, suggesting that loss of *Men1* increases accumulation of DNA damage in a ROS-independent manner.

Q11: Effects of Men1 deficiency on cell cycle checkpoint responses should be assessed as a possible means for the DNA damage noted. Figure 5 -Cell cycle alterations in cells with or without *Men1* should be included.

Response: We appreciate the insightful suggestions from the reviewer. According to the Reviewer's suggestion, *Men1^{fl/fl}* and *Men1^{Δ/Δ}* MEF cells were exposure to 3 Gy of IR, and collected one hour after IR. Western blot displayed that deletion of *Men1* distinctly activated phosphorylation of cell cycle checkpoints p-CHK1 and p-CHK2 induced by IR exposure in MEF cells (Supplementary Fig. 5g). A549-sh*MEN1* and A549-sh*Luc* cells were serum-starved for 24 hours and collected 8 hours after release from serum starvation and stained with propidium iodide (PI). Flow cytometer results showed that 24.03% of the A549-sh*Luc* cells progressed from G0/G1 to S phase and 35.8% of A549-sh*MEN1* cells entered S phase 8 hours after release from serum starvation (data not shown in the manuscripts, below). Our data results demonstrate that loss of menin accelerates progression from G0/G1 to S phase. It has been reported that deletion of *Men1* in MEF cells accelerates G0/G1 to S phase entry through increased cyclin-dependent kinase 2 (CDK2) activity as well as decreased expression of CDK inhibitors *p18^{Ink4c}* and *p27^{Kip1}* (1). Moreover, menin-mediated repression of cyclin B2 is crucial for inhibiting G2/M transition and cell proliferation (2). These results indicate that menin regulates DNA damage repair at least partly through controlling cell cycle checkpoints.

Q12 Figure 6: Panel B-E, inclusion of standard error and P values are needed.

Response: We fully agree with the reviewer's comment. According to reviewer's suggestion, the IHC quantification and statistical analysis were supplemented. For automatic quantification, digital images of stained tissues were captured with a Motic microscope using DigiLabII-Client software. Quantification of IHC staining was calculated by image pro plus software. The same index for the same batch of samples uses the unified parameters. All the indexes were measured by positive area, the parameter as (H:

0-30, S: 0-180, I: 0-180 or H: 0-20, S: 0-160, I: 0-160). For automatic quantification the area of positive cells per sample, we randomly analyzed three fields (400 times magnification, 800x600 pixel) per individual sample, then calculated the mean value for each sample. The mean value and stand error of mean were used to calculate for each group. At last, the p-value was calculated by Two-tailed unpaired t test.

The IHC results clearly showed that menin was stained in the nucleus and that menin expression in tumors, compared to that in adjacent lung tissues, was markedly reduced or undetectable in 42 out of the 157 NSCLC cases (26.8%) and in 15 out of the 34 SCLC cases (44.1%). Compared with those from NSCLC patients, samples from SCLC patients possessed significantly strong NE profiles and exhibited reduced mesenchymal characteristics. Furthermore, the menin-low (-) patients had significantly high staining for NE markers (NCAM1, NSE) but low staining for Vimentin compared with menin-high (+) patients. The correction has been made in Fig.6b,c,d.

Q13 Figure 7-Panel C. Statistics should be included, where are p values?

Response: According to the reviewer's suggestion, we have performed Two-tailed unpaired t test statistical analysis with the SPSS statistical software. The statistical results were indicated in the Fig. 7c.

Q14 Figure 7-Panels G, H. Can these be quantified?

Response: According to the reviewer's suggestion, the IHC quantification and statistical analysis were presented in Supplementary Fig. 7g,h.

References

1. Schnepf, R.W *et al.* Mutation of tumor suppressor gene Men1 acutely enhances proliferation of pancreatic islet cells. *Cancer Res* **66**, 5707-5715 (2006).
2. Wu, T *et al.* Regulation of cyclin B2 expression and cell cycle G2/m transition by menin. *J Biol Chem* **285**, 18291-1300 (2010).

POINT-BY-POINT RESPONSE TO THE REVIEWERS 3' COMMENTS:

The authors use a genetically engineered mouse model (GEMM) to study lung cancer in mice induced with a *KRAS* mutation and loss of function of *MEN1*. They find that combined mutations lead to increase in lung cancers, increase in tumors with neuroendocrine (NE) differentiation, decreased TP53 and Rb status in these NE tumors, increased accumulated DNA damage after radiation, and anti-tumor responses to a LSD1 inhibitor. They also examine human lung tumors for *MEN1* expression level and find lower *MEN1* is associated with impaired survival. Selected genetic functional studies are done in two human non-small cell lung cancer (NSCLC) lines and mouse embryonic fibroblasts (MEFs). They conclude: "Our discoveries strongly suggest that epigenetic targeting of LSD1 is an effective strategy for the treatment of menin low NE-type lung cancer. In addition, our findings suggest the potential application of LSD1 inhibitors for the treatment of NE-type neoplasms originating from multiple organs. The present study defines an interesting epigenetic mechanism by which menin controls senescence and differentiation during lung carcinogenesis and provides a novel GEMM for further investigating the pathogenesis of P-NETs in vivo.

Comments to the Authors:

The studies are technically well done and clearly presented. However, with relevance to publication as a Nature Communication the central issues of this report are: 1. Do they provide substantial new information for a major role of *MEN1* in NE lung cancers (even for a subset of these); 2. Do they provide a mechanistic understanding of such a role; 3. Have they identified new targeted therapy for lung cancers with loss of *MEN1* function?

I believe they have provided some new mechanistic understanding for a potential *MEN1* role - mainly the effect on TP53, RB and overcoming *KRAS* oncogene induced senescence, increase in DNA damage.

Q1: However, they do not provide convincing data that *MEN1* loss of function plays a major role in NE cancers of any type, and they do not have data that using LSD1 inhibitors have major importance in preclinical model responses for *MEN1* loss of function human NE tumors.

Response: We appreciate the insightful suggestions from the reviewer. It has been reported that somatic inactivating mutations of *MEN1* alleles are found in various neuroendocrine tumors, including pancreatic neuroendocrine tumors (1), sporadic gastrinomas (2), and parathyroid tumors (3). High frequencies of *MEN1* mutation was positively correlated with upregulation of the master neuroendocrine regulator ASCL1 in gastrointestinal stromal tumor (4). Moreover, 44% of pancreatic neuroendocrine tumors had somatic inactivating mutations in *MEN1* gene, but not in pancreatic ductal adenocarcinomas (5). Importantly, the somatic inactivating mutations of *MEN1* alleles were found in 4% of large-cell neuroendocrine carcinoma; 3.7% of large-cell neuroendocrine carcinoma have a loss of heterozygosity of *MEN1* (6). These findings support a genetic similarity with neuroendocrine tumors in *MEN1* mutation. *Men1*^{+/-} inactivating mutation-driven genetically engineered mouse models accurately reflect the biology of various neuroendocrine tumors, including pancreatic islet, gastric neuroendocrine tumor and lung adenocarcinoma (7). *Men1*^{+/-}-induced lung tumors developed high penetrance and neuroendocrine characteristics (8). In the present study, we found that *Men1* deficiency dramatically accelerates *Kras*-induced lung carcinogenesis and induces the differentiation of epithelial cells into NE cells. These findings demonstrated that *MEN1* inactivating mutation comprehensively participated in the development of diverse neuroendocrine tumors.

We further evaluated the effectiveness of SP2509 treatment *in vitro* and *in vivo*. The results from clone formation experiment indicated that exposure of SP2509 profoundly reduced the number of proliferative cell clones in the A549-sh*MEN1* cells relative to A549-sh*Luc* cells (data not shown in the manuscripts, below). Consistent with *in vitro* data, treatment of SP2509 significantly reduced tumor volume and weight in human A549-sh*MEN1* xenograft lung cancer (see response to Q3). Furthermore, the Kaplan-Meier survival curves showed that KMS-SP2509 mice experienced long-term survival than KMS-vehicle mice during observation period (Supplementary Fig. 7e). These data indicate an effective anti-tumor effects of SP2509 for suppressing lung cancers with *MEN1* inactivation.

Q2: The major issue is what is the frequency of *MEN1* loss in lung cancers, NE lung cancers, and *KRAS* mutant lung cancers? The available deposited data indicate these subsets must be very uncommon. While the authors should have presented this information, one can go into important databases and discover: there are no *MEN1* mutations in the ~90 small cell lung cancers (SCLCs) in the George data set; 6 *MEN1* mutations are present in lung adenocarcinomas (>600) in the TCGA dataset-3 of which have a *KRAS* mutation; 3 *MEN1* mutations in squamous lung cancers (>500) none with *KRAS* mutations; and searching deposited mutation datasets on human lung cancer lines there are very few *MEN1* mutations found-but such SCLC and NSCLC preclinical models that can be tested do exist. Particularly since the authors focused on a *KRAS* based GEMM for their studies, the combination of *KRAS* and *MEN1* inactivation in NSCLC are essentially rare.

Response: We appreciate the insightful suggestions from the reviewer. The statistical analysis of TCGA dataset indicate that frequency of *MEN1* mutation in SCLC (0%), large-cell neuroendocrine carcinoma (4%), atypical lung carcinoid (20%), and typical lung carcinoid (6%), and *KRAS* mutant lung cancers (1.4%) and breast cancers (1.7%) is very rare. The 23% of *KRAS* mutation (n=222 cases) and 1% of *MEN1* mutation (n=10 cases) were found in 1,144 lung cancer; 3 of 10 *MEN1* mutant lung adenocarcinomas combined with *Kras* mutations. The data indicate that subsets of *MEN1/Kras* mutant lung cancer is uncommon genetic events.

However, our previous reports (9) found that reduced menin expression is associated with the activation of *Kras* ($p < 0.05$) in primary human lung cancer. The inactivation of *MEN1* is associated with DNA methylation at the *MEN1* promoter by *Kras*. The activated *Kras* up-regulates the expression of DNA methyltransferases (DNMTs) and enhances the binding of DNMT1 to the *MEN1* promoter, leading to increased DNA methylation at the *MEN1* gene in lung cancer. On the other hand, menin reduces the level of Ras-GTP at least partly by preventing GRB2 and SOS1 from binding to *Kras*. This finding suggests that the inactivation of menin expression in lung cancer possibly associated with epigenetic regulation, but not genetic mutation. More recently, our unpublished data from whole exon sequencing found that lung tumors developed *KRAS*^{G12D} mutation in *Men1* deletion-induced spontaneous lung cancer mouse model (33.3% of mutation rate). This finding suggests that the *Kras* mutation was preferentially affected by altered DNA damage response by inactivation of *Men1*. Together, these findings uncover a previously unknown link between activated K-Ras and menin, an important interplay governing tumor activation and suppression in the development of lung cancer.

Q3: In any event, for the therapeutic aspects with an LSD1 inhibitor we need to see effects of such treatments on human preclinical models with patient acquired *KRAS* mutations and loss of *MEN1* function both in vitro and in vivo (xenografts).

Response: We would like to thank the reviewer for their precious advice. To delineate the role of SP2509 in lung cancer growth *in vivo*, A549-sh*Luc* and A549-sh*MEN1* cells were subcutaneously transplanted into nude mice. Treatment of SP2509 significantly suppressed A549-sh*MEN1* tumor volume and weight *in vivo* but did not block A549-sh*Luc* xenografts at 19 day. These studies thus offer a new therapeutic strategy for menin-deficient NE-type lung cancer by restoring senescence. The correction has been made in Results section (page 15), and now the data is presented in Supplementary Fig. 7h,i.

References

1. Scarpa, A *et al.* Whole-genome landscape of pancreatic neuroendocrine tumours. *Nature* **543**, 65-71 (2017).
2. Zhuang, Z *et al.* Somatic mutations of the *MEN1* tumor suppressor gene in sporadic gastrinomas and insulinomas. *Cancer Res* **57**, 4682-4686 (1997).
3. Heppner, C *et al.* Somatic mutation of the *MEN1* gene in parathyroid tumors. *Nat Genet* **16**, 375-8 (1997).
4. Pantaleo, M. *et al.* Genome-wide Analyses Identifies *MEN1* and *MAX* Mutations and a Neuroendocrine-like Molecular Heterogeneity in Quadruple WT GIST. *J Molecular Cancer Res* **15**, 553 (2017).
5. Jiao, Y *et al.* *DAXX/ATRX*, *MEN1* and *mTOR* pathway genes are frequently altered in pancreatic neuroendocrine tumors. *Science* **331**, 1199-203 (2011).
6. Simbolo, M *et al.* Lung neuroendocrine tumours: deep sequencing of the four World Health Organization histotypes reveals chromatin-remodelling genes as major players and a prognostic role for *TERT*, *RB1*, *MEN1* and *KMT2D*. *J Pathol* **241**, 488-500 (2017).
7. Crabtree, J.S *et al.* A mouse model of multiple endocrine neoplasia, type 1, develops multiple endocrine tumors. *Proc Natl Acad Sci U S A* **98**, 1118-1123 (2001).
8. Pei, X.H *et al.* *p18^{Ink4c}* collaborates with *Men1* to constrain lung stem cell expansion and suppress non-small-cell lung cancers. *Cancer Res* **67**, 3162-3171(2007).
9. Wu, Y *et al.* Interplay between menin and K-Ras in regulating lung adenocarcinoma. *J Biol Chem* **287**, 40003-11 (2012).

Reviewers' comments:

Reviewer #1 (Remarks to the Author):

Although the authors made an effort to provide new data, I am not convinced in general by the new experiments. They represent an improvement over the initial submission but still fall short to convincingly demonstrate most of their claims. This is for example particularly exemplified by the IHC analyses. Images and quantifications are not very convincing and some of the stainings show weird positivity that do not fit well with previously published data. The analysis of cellular senescence is very poor. They only added some Ki67 stainings on consecutive sections and not extra markers and never on the same slide. Cell culture analysis is always done in bulk and never on an individual cell basis.

In summary, I don't think the manuscript is robustly supported by clear experimental evidence.

Reviewer #2 (Remarks to the Author):

The authors of this paper have made an extraordinary attempt to improve this paper. While they have still not shown some mechanistic links between MEN1 loss and DNA damage and repair, the links are very interesting and I now think that this paper is ready for publication. I appreciated all of the efforts that these authors made to satisfy reviewers concerns.

Reviewer #3 (Remarks to the Author):

NCOMMS-19-07700A

MEN1 deficiency leads to neuroendocrine differentiation of lung cancer by disrupting the DNA damage response and senescence progression

The authors have prepared a detailed rebuttal to all of the reviewers' comments including providing additional experimental details and a lot of quantification of various image and other analyses. At the end of the day the question is have they provided us with important new information on neuroendocrine lung cancer (or other cancers) relevant to their study of loss of MEN1 function, particularly information that could lead to new targeted therapy of these aggressive tumors? I think I am now convinced they have discovered something new and important. Part of the problem is in their effort to show as much data as possible in their multi panel figures (main and supplemental) they have made some of these key findings difficult to extract. This was the third time I went over their paper in detail. The first two were with the initial review, and if I hadn't spent more than an hour this third time trying to see if they had discovered something I would have still missed it in the revised version.

1. Basically here is the issue: they start with mouse genetically engineered models with MEN1 deficiency and make all of their findings concerning with neuroendocrine differentiation, TP53 and RB inactivation, ASCL1 upregulation and role in KRAS oncogene induced senescence. All of this is good and in a mouse model.

2. They then study MEN1 protein expression in clinically annotated human non-small cell lung cancer (NSCLC) and small cell lung cancer (SCLC) samples. In these there they claim loss of MEN1 protein expression in 27% of NSCLC cases and 44% of SCLC cases. The quantitative data supposedly supporting that statement is in panel 6 b and an example in 6a. What they need to show is that they really have many NSCLC cases with loss of MEN1 expression quantitatively (they show MEN+ area not tumors with MEN negative areas) – it is that that is key. Interesting that SCLC that are neuroendocrine positive with either MEN1 positive or negative represent a different case and it would be interesting to know about SCLCs with high MEN1 protein expression for other key things like NE differentiation -but that is a side light.

3. So the first thing is to document there truly are a substantial number of NSCLCs that don't express MEN1 protein. Now from mutation analyses we know that this will rarely be due to MEN1

mutations. So, there must be epigenetic inactivation. While I would have liked to know about this mechanism and whether there was also allele loss at the MEN1 locus that is also a side light.

4. What is further key is the study of the individual MEN1 negative NSCLCs and demonstration in each of them of turn on of neuroendocrine function. We know that only 5-10% of otherwise classified NSCLCs express neuroendocrine markers so it would be important to know about what happens in these 27% NSCLCs with supposed loss of MEN1 expression. Is there really a much larger pool of NSCLCs with neuroendocrine features we have been missing? Or are their NSCLCs with MEN1 loss of expression some of which are neuroendocrine and some of which are not?

5. Also since they put such an emphasis on MEN1 loss of expression with KRAS mutation, what is the connection between human NSCLCs that don't express MEN1 and KRAS mutation? Are they tightly linked or not? Whatever the answer is we need to know.

6. Then they treat their GEMM with the LSD1 inhibitor. The key data are in fig 7f. However, there is no way to know from what is presented did they really get dramatic responses or long-term survival.

7. Then they test the LSD1 inhibitor in the A549 model comparing MEN1 knockdown to control in a xenograft study. The problem with this, is that A549 did not arise with loss of MEN1 function so we don't know what the response to the LSD1 inhibitor in this context means. What we need is to see a NSCLC xenograft that lacked MEN1 expression in its pathogenesis – how does it respond? Clearly any clinical trial would screen NSCLC patients for loss of MEN1 protein expression and then put them on the LSD1 inhibitor. Of course, it would be important to know the KRAS mutation and neuroendocrine expression status of such a tumor.

Finally, in the abstract and the Discussion it would be useful to have these ideas clearly stated. While they might be clear to the authors, it took me digging through this to get this straight. I am someone who has spent my life trying to find such connections and examples, and I almost rejected the rebuttal version, since I thought I had expressed most of these key issues in the original review.

POINT-BY-POINT RESPONSE TO THE REVIEWERS 1' COMMENTS:

Although the authors made an effort to provide new data, I am not convinced in general by the new experiments. They represent an improvement over the initial submission but still fall short to convincingly demonstrate most of their claims. This is for example particularly exemplified by the IHC analyses. Images and quantifications are not very convincing and some of the staining show weird positivity that do not fit well with previously published data. The analysis of cellular senescence is very poor. They only added some Ki67 staining on consecutive sections and not extra markers and never on the same slide. Cell culture analysis is always done in bulk and never on an individual cell basis. In summary, I don't think the manuscript is robustly supported by clear experimental evidence.

Response: We are grateful to the reviewer for the valuable comments on this manuscript. In order to improve the quality of manuscript, we carefully checked all the IHC data and images, and re-performed some of the IHC staining, particularly for p53 and p16 which were referred to previous reports (1,2). The IHC quantification was referred to another study (3). In addition, IHC quantitative software, raw data and operational procedures of quantitative software have been uploaded. In the revised manuscript, we replaced some poor-quality IHC pictures, which were shown in the Fig. 5a and Fig. 7g. We sincerely hope that these corrections are helpful to improve the quality of the manuscript.

The reviewer kindly mentioned that "they only added Ki67 staining in the revised manuscript". More convincing staining will prove the conclusion that loss of *Men1* inhibits senescence. We fully agree with the suggestions by the reviewer, and the trimethylation at Lys9 of histone 3 (H3K9me3), a marker of senescence-associated heterochromatic foci (SAHF) (4,5), was used to confirm the senescence. In the current revised manuscript, we re-performed IHC staining for p16 and H3K9me3 in the lung consecutive sections of KS or KMS mice treated with or without SP2509. The results showed that the staining of H3K9me3 and p16 were significantly higher in KMS-SP2509 mice lung tissues relative to KMS-vehicle mice. As cellular senescence is based on a stable cell cycle arrest, and absence of proliferative markers, such as Ki67 protein, is an essential phenotype to document senescence (6). Indeed, the KMS mice have a stronger Ki67 staining than that of KS mice lung tumor tissues. Moreover, the levels of Ki67 in the KMS-SP2509 mice significantly lower than that of the KMS-vehicle mice, which were shown in Fig. 5a,b and Fig. 7g. These results indicate that loss of *Men1* suppresses cellular senescence and promotes cell growth in lung cancer.

Finally, the reviewer has mentioned that “Cell culture analysis is always done in bulk and never on an individual cell basis”. This is a very constructive suggestion for fully proving the conclusion that menin is required for cellular senescence. To rule out the possible effect of high cell density on senescence, we inoculated the primary *Men1^{fl/fl}* and *Men1^{Δ/Δ}* MEF cells at a density of 40-50%, and immunofluorescence was performed with Ki67 and p21 (a cell cycle inhibitor) antibodies at 3 days after exposure to 3 Gy IR. We observed that loss of *Men1* notably increased Ki67 and reduced p21 staining in the *Men1^{Δ/Δ}* MEF cells with or without IR treatment compared with the *Men1^{fl/fl}* MEF cells, as shown in following figure. This result further supports our conclusion that *Men1* deficiency inhibits cellular senescence and promotes proliferation.

We believe and we wish our reviewer also agree with us that these improvements are very helpful to solidify our conclusion. Thank you again for your pertinent comments.

References

1. Cottage, C.T. *et al.* Targeting p16-induced senescence prevents cigarette smoke-induced emphysema by promoting IGF1/Akt1 signaling in mice. *Commun Biol* **2**, 307-307 (2019).
2. Qi, H.G. *et al.* Tea polyphenols prevent lung from preneoplastic lesions and effect p53 and bcl-2 gene expression in rat lung tissues. *Int J Clin Exp Pathol* **6**, 1523-1531 (2013).
3. Fang, Y. *et al.* CD36 inhibits β -catenin/c-myc-mediated glycolysis through ubiquitination of GPC4 to repress colorectal tumorigenesis. *Nature Commun* **10**, 3981 (2019).
4. Narita, M. *et al.* Rb-mediated heterochromatin formation and silencing of E2F target genes during cellular senescence. *Cell* **113**, 703–716 (2003).
5. Yu, Y. *et al.* Targeting the senescence-overriding cooperative activity of structurally unrelated H3K9 demethylases in melanoma. *Cancer Cell* **33**, 322-336 (2018).
6. Dewberry, R.M. *et al.* Interleukin-1 receptor antagonist (IL-1RN) genotype modulates

the replicative capacity of human endothelial cells. *Circ Res* **92**, 1285-1287 (2003).

POINT-BY-POINT RESPONSE TO THE REVIEWERS 2' COMMENTS:

The authors of this paper have made an extraordinary attempt to improve this paper. While they have still not shown some mechanistic links between *MEN1* loss and DNA damage and repair, the links are very interesting and I now think that this paper is ready for publication. I appreciated all of the efforts that these authors made to satisfy reviewers concerns.

Response: We sincerely appreciate the reviewer's positive comments and the suggestion for publication of this work. We also interested in the mechanistic links between *MEN1* and DNA damage repair. Recently, we have found some interesting phenomena, such as menin was recruited to the DSB sites, promoted HR and inhibited NHEJ repair efficiency. In addition, we also observed that menin/MLL complex mediated chromatin H3K4me3 modification at DSB sites and altered chromatin conformation to regulate DNA damage response. We will further elucidate the mechanisms by which menin regulates DNA damage repair, which is of great significance for cancer therapeutics and tumorigenesis mechanisms induced by *MEN1* deletion. Thanks again for the reviewer's pertinent comments on our paper.

POINT-BY-POINT RESPONSE TO THE REVIEWERS 3' COMMENTS:

The authors have prepared a detailed rebuttal to all of the reviewers' comments including providing additional experimental details and a lot of quantification of various image and other analyses. At the end of the day the question is have they provided us with important new information on neuroendocrine lung cancer (or other cancers) relevant to their study of loss of MEN1 function, particularly information that could lead to new targeted therapy of these aggressive tumors? I think I am now convinced they have discovered something new and important. Part of the problem is in their effort to show as much data as possible in their multi panel figures (main and supplemental) they have made some of these key findings difficult to extract. This was the third time I went over their paper in detail. The first two were with the initial review, and if I hadn't spent more than an hour this third time trying to see if they had discovered something I would have still missed it in the revised version.

1. Basically here is the issue: they start with mouse genetically engineered models with MEN1 deficiency and make all of their findings concerning with neuroendocrine differentiation, TP53 and RB inactivation, ASCL1 upregulation and role in KRAS oncogene induced senescence. All of this is good and in a mouse model.

Q1: 2. They then study MEN1 protein expression in clinically annotated human non-small cell lung cancer (NSCLC) and small cell lung cancer (SCLC) samples. In these there they claim loss of MEN1 protein expression in 27% of NSCLC cases and 44% of SCLC cases. The quantitative data supposedly supporting that statement is in panel 6 b and an example in 6a. What they need to show is that they really have many NSCLC cases with loss of *MEN1* expression quantitatively (they show MEN+ area not tumors with MEN negative areas)-it is that that is key. Interesting that SCLC that are neuroendocrine positive with either MEN1 positive or negative represent a different case and it would be interesting to know about SCLCs with high MEN1 protein expression for other key things like NE differentiation-but that is a side light.

Q2: 4. What is further key is the study of the individual MEN1 negative NSCLCs and demonstration in each of them of turn on of neuroendocrine function. We know that only 5-10% of otherwise classified NSCLCs express NE markers so it would be important to know about what happens in these 27% NSCLCs with supposed loss of MEN1 expression. Is there really a much larger pool of NSCLCs with neuroendocrine features we have been missing? Or are their NSCLCs with MEN1 loss of expression some of which are neuroendocrine and some of which are not?

Response: We are grateful to our reviewer for the constructive comments and suggestions on our manuscript. We observed that loss of *Men1* enhances the expression of NE markers such as NCAM1, NSE, CgA and Syn, and we confirmed the closely correlation between loss of menin and NE differentiation of lung cancer in **GEMMs**. In the **clinical samples**, NSCLC and SCLC were respectively divided into menin-low and -high expression groups by comparing with precancerous tissues using IHC staining by double blind method. We apologize that the previous description was not accurate and clear. Here, we modified menin (-) and menin (+) to menin-low and -high in the revised manuscript (Figure 6 in the manuscript).

We are honored to have an opportunity to discuss with the reviewer about the interesting albeit controversial issues on the optimal means of identifying NE differentiation in clinical. According to the WHO classification, Ki67 proliferative index > 20% is the cut-off level for neuroendocrine carcinoma (1). At present, the identifications for NE differentiation in non-small cell lung cancer (NSCLC) is based on immunohistochemical (IHC), ultrastructural, and serological data (2,3). Clinically, using single molecular marker is not adequate for diagnosis NE differentiation. Combination of multiple indicators, such as CgA, Syn, NSE, and NCAM, is commonly used in clinical practice (4). It has been reported that the positivity for either NE marker (CgA, Syn, or NCAM) was identified in 13.6% of NSCLC (76/558) using tissue microarray (TMA) and IHC (5). Here, we detected that reactivity for NCAM1 in 19.2% and for NSE in 25% of 157 NSCLCS cases, respectively. It was suggested that the variable proportion of NSCLC NE differentiation may be influenced by the usage of different markers or regional disparity. It seems that the accurate and effective molecular markers to identify NE-type lung cancer are still lacking. Therefore, a better understanding of the molecular mechanisms will provide more meaningful biomarkers, and thereby further improving the accuracy of clinical diagnostic of pNET.

In the present study, we detected high expression of NE markers including CgA, Syn, NSE, and NCAM1 in the lung tumor tissues of mice with *Men1* deficiency. Statistically,

menin was negatively correlated with Syn (pearson'R =0.562, p=0.0012) and CgA expression (pearson'R = 0.585, p=0.0005) in the GEMM. Similar results were observed in 191 lung cancer patient samples that menin was negatively correlated with NCAM1 (pearson'R = -0.265, p=0.0449) and NSE expression (pearson'R = -0.432, p=0.0005), respectively. Our results indicate a strong statistical correlation between menin expression and NE differentiation in lung cancer. As the reviewer stated, we also noticed that some SCLCs showed high expression of menin, suggesting that there are alternative pathways for controlling NE differentiation in *MEN1*-independent manner. Currently, the mechanisms of NE differentiation in lung tumor remains unclear. It was reported that some genes including *p53*, *Rb*, *LRP1B*, *KMT2D*, *CSMD3*, and *MYC* have a somatic functional mutation, which are important events in NE differentiation of lung cancer (6,7). For example, deletion of *p53/Rb* produces tumors with SCLC features in GEMM, which is a typical pNET model (8). An amplification of *MYC* also drives progression of SCLC, which is a variant NE subtype with high expression of *NEUROD1* (9). These findings suggested that multiple molecular mechanisms are involved in the pathogenesis of pNET. We conclude that *MEN1* gene is one of the key regulators for NE differentiation of lung cancer, which is at least partly dependent on *p53/Rb*, rather than *NEUROD1* or *MYC* pathways.

Q3: 3. So the first thing is to document there truly are a substantial number of NSCLCs that don't express *MEN1* protein. Now from mutation analyses we know that this will rarely be due to *MEN1* mutations. So, there must be epigenetic inactivation. While I would have liked to know about this mechanism and whether there was also allele loss at the *MEN1* locus that is also a side light.

Response: High frequencies of *MEN1* somatic inactivating mutations are found in various neuroendocrine tumors, including 41- 44% pancreatic neuroendocrine tumors (10,11), 33% sporadic gastrinomas and 17% insulinomas (12). Mechanistically, the mutant menin, but not wild-type menin, interacts with both the molecular chaperone Hsp70 and the Hsp70-associated ubiquitin ligase CHIP, further promotes the ubiquitination and degradation of the mutant menin (13,14). Importantly, the somatic inactivating mutations of *MEN1* alleles were also found in pNET, such as 4% of large-cell neuroendocrine carcinoma, 6% typical lung carcinoid, and 20% atypical lung carcinoid, respectively (6). In addition, we also noticed the reports that a heterozygous deletion of the *MEN1* gene in 15.1% of typical lung carcinoid and 22.9% of atypical lung cancers (6). It suggests that genetic event is one of the mechanisms for silencing expression of menin in NE-type lung tumors.

We explored the mechanisms of inactivation of menin expression in NSCLC. We previously found one single nucleotide polymorphism (C to T change at codon 7), with no change in amino acid sequence (15). TCGA dataset showed that only 1-5% of lung adenocarcinomas had *MEN1* mutations. These findings indicated that the reduction of menin by *MEN1* genetic mutation is a rare event in human lung adenocarcinoma, so it may depend on epigenetic mechanisms. Indeed, we previously reported that the inactivation of menin is associated with increased DNA methylation at the CpG sites of *MEN1* promoter by KRAS in lung cancer cells, and the methylation of the *MEN1* promoter region in primary lung adenocarcinoma tissues was significantly increased compared with adjacent tissues (15). These observations indicate that the reduction of menin expression in NSCLC is predominantly caused by epigenetic mechanisms.

Q4: 5. Also since they put such an emphasis on MEN1 loss of expression with KRAS mutation, what is the connection between human NSCLCs that don't express MEN1 and KRAS mutation? Are they tightly linked or not? Whatever the answer is we need to know.

Response: We have unraveled a novel feedback loop of KRAS and *MEN1* in human NSCLC. KRAS inhibits menin expression by promoting DNMT1 binding to the *MEN1* promoter and increased DNA methylation, whereas menin inhibits KRAS-mediated signaling via suppressing SOS1-mediated activation of RAS by blocking GRB2-SOS1 from binding to RAS (15). We reported that DNA methylation of *MEN1* promoter was increased in lung cancer tissues as compared with the adjacent tissue, and the *MEN1* mRNA level was up-regulated by DNA methylation inhibitor in lung cancer cells. In turn, methylation of *MEN1* promoter was decreased in KRAS knockdown (KD) A549 cells. These observations were consistent with the notion that KRAS may repress menin expression via increasing *MEN1* methylation. Supporting this notion, we demonstrated that the expression of DNMT1 and DNMT3B was substantially reduced in KRAS KD A549 cells. CHIP assay results provided a direct evidence that KRAS can promote DNMT1 binding to the *MEN1* promoter. Intriguingly, we also found that menin reduced the active RAS-GTP form in A549 cells. During investigating the mechanism whereby menin inhibits RAS activity, we found that ectopic expression of menin in A549 cells decreased binding of SOS1 and GRB2 to RAS, without affecting the expression of GAP, SOS1 and GRB2. Menin bound to endogenous GRB2 and SOS1. Thus, it is likely that menin binds to the GRB2-SOS1 complex, reduces the binding of SOS1 and GRB2 to RAS, which results in the decreased RAS-GTP. While RAS mutations at 12, 13 and 61 positions impair the intrinsic GTPase activity, causing the accumulation of RAS-GTP, menin-mediated repression of SOS1 binding to K-RAS may still reduce KRAS-GFP levels. Our findings

uncover a new mechanism whereby menin represses the KRAS through inhibiting the formation of GRB2-SOS1-RAS complex, thereby inhibited the RAS/RAF/MEK/MAPK pathway in the formation of lung cancer.

We uncovered a tightly link between activated KRAS and menin inactivation, an important interplay governing tumor activation and suppression in the development of lung cancer. Here, we further propose that *MEN1* and *KRAS* cooperatively control the directional differentiation of epithelial cell-derived lung cancer cells.

Q5: 6. Then they treat their GEMM with the LSD1 inhibitor. The key data are in fig 7f. However, there is no way to know from what is presented did they really get dramatic responses or long-term survival.

Response: We thank the reviewer for pointing out this aspect. The Kaplan-Meier survival curves showed that treatment of SP2509 significantly prolonged the median survival from 42 to 74 days in KMS mice. The data is presented in Supplementary Fig. 7f.

Q6: 7. Then they test the LSD1 inhibitor in the A549 model comparing *MEN1* knockdown to control in a xenograft study. The problem with this, is that A549 did not arise with loss of *MEN1* function so we don't know what the response to the LSD1 inhibitor in this context means. What we need is to see a NSCLC xenograft that lacked *MEN1* expression in its pathogenesis-how does it respond? Clearly any clinical trial would screen NSCLC patients for loss of *MEN1* protein expression and then put them on the LSD1 inhibitor. Of course, it would be important to know the KRAS mutation and neuroendocrine expression status of such a tumor.

Response: We have also noticed this problem that A549-sh*MEN1* cells did not obviously promote tumor growth comparing with A549-sh*Luc* cells at 19 days. Initially, we designed the experiment into four groups, 10 nude mice in each group. The tumor volume in the A549 sh*MEN1*-vehicle was significantly larger than that in the A549 sh*Luc*-vehicle in the 9 to 15 days. However, three nude mice with large tumors died spontaneously during the observation in the sh*MEN1*-vehicle group. So, statistical analysis showed that there was no statistical difference in tumor volume between the A549 sh*MEN1*-vehicle and A549 sh*Luc*-vehicle groups (p=0.0663) at 19 days.

Reviewer mentioned that "What we need is to see a NSCLC xenograft that lacked

MEN1 expression in its pathogenesis-how does it respond?" Firstly, sh*MEN1*-mediated knockdown (KD) of menin was verified by Western blot and menin depletion in the A549 sh*MEN1* cells was >90% relative to A549 sh*Luc* cells. IHC staining also displayed that expression of menin was obviously reduced in the A549 sh*MEN1* xenograft tumors relative to A549 sh*Luc* xenograft tumors. Pathologically, the findings in figure 5 have confirmed that *MEN1* was required for *KRAS*^{G12D} activation-induced senescence. Therefore, we hypothesized that SP2509 improves the malignant phenotypes of lung adenocarcinoma via restoring the inhibition of senescence by *MEN1* deletion. IHC of Ki67 (a proliferation marker) and H3K9me3 (a cellular senescence marker) in all tumor samples were performed. Moreover, the KD of *MEN1* significantly increased Ki67 index, decreased p16 and H3K9me3 staining in the A549 sh*MEN1*-vehicle compared with A549 sh*Luc*-vehicle xenograft tumors. Importantly, a significantly reduced Ki67 staining and increased p16 and H3K9me3 staining were observed in the A549 sh*MEN1*-SP2509 relative to A549 sh*MEN1*-vehicle xenograft tumors. Similar results were not observed between the A549 sh*Luc*-vehicle and A549 sh*Luc*-SP2509 xenograft tumors (Supplementary Fig. 7i,j). Our results demonstrate that SP2509 predominantly inhibits the growth of lung tumors with menin low expression.

Q7: Finally, in the abstract and the discussion it would be useful to have these ideas clearly stated. While they might be clear to the authors, it took me digging through this to get this straight. I am someone who has spent my life trying to find such connections and examples, and I almost rejected the rebuttal version, since I thought I had expressed most of these key issues in the original review.

Response: We apologize for the confusion caused by the unconcise introduction and discussion. During revision, we have deleted some unnecessary descriptions, and rearranged our manuscript. These modifications were marked in red in the manuscript. We sincerely hope that these modifications will be helpful to improve the quality of our manuscript.

References

1. Mengoli, M.C. *et al.* The 2015 World Health Organization Classification of lung tumors: new entities since the 2004 Classification. *Pathologica* **110**, 39-67(2018).
2. Abbona, G. *et al.* Chromogranin A gene expression in non-small cell lung carcinomas. *J Pathol* **186**, 151-156 (1998).
3. Pericleous M. *et al.* Well-differentiated bronchial neuroendocrine tumors: Clinical management and outcomes in 105 patients. *Clin Respir J* **12**, 904-914 (2018).

4. Howe, M.C. *et al.* Neuroendocrine differentiation in non-small cell lung cancer and its relation to prognosis and therapy. *Histopathology* **46**, 195-201 (2005).
5. Ionescu, D.N. *et al.* Non-small cell lung carcinoma with neuroendocrine differentiation-an entity of no clinical or prognostic significance. *Am J Surg Pathol* **31**, 26-32 (2007).
6. Rickman, D.S. *et al.* Biology and evolution of poorly differentiated neuroendocrine tumors. *Nat Med* **23**, 1-10 (2017).
7. Simbolo, M. *et al.* Lung neuroendocrine tumours: deep sequencing of the four World Health Organization histotypes reveals chromatin-remodelling genes as major players and a prognostic role for TERT, RB1, MEN1 and KMT2D. *J Pathol* **241**, 488-500 (2017).
8. Meuwissen, R. *et al.* Induction of small cell lung cancer by somatic inactivation of both Trp53 and Rb1 in a conditional mouse model. *Cancer Cell* **4**, 181-189 (2003).
9. Mollaoglu, G. *et al.* MYC Drives Progression of Small Cell Lung Cancer to a Variant Neuroendocrine Subtype with Vulnerability to Aurora Kinase Inhibition. *Cancer Cell* **31**, 270-285 (2017).
10. Scarpa, A *et al.* Whole-genome landscape of pancreatic neuroendocrine tumours. *Nature* **543**, 65-71 (2017).
11. Jiao, Y. *et al.* DAXX/ATRX, MEN1, and mTOR pathway genes are frequently altered in pancreatic neuroendocrine tumors. *Science* **331**, 1199-203 (2011).
12. Zhuang, Z *et al.* Somatic mutations of the *MEN1* tumor suppressor gene in sporadic gastrinomas and insulinomas. *Cancer Res* **57**, 4682-4686 (1997).
13. Shimazu, S. *et al.* Correlation of mutant menin stability with clinical expression of multiple endocrine neoplasia type 1 and its incomplete forms. *Cancer Sci* **102**, 2097-2102 (2011).
14. Yaguchi, H. *et al.* Menin missense mutants associated with multiple endocrine neoplasia type 1 are rapidly degraded via the ubiquitin-proteasome pathway. *Mol Cell Biol* **24**, 6569-6580 (2004).
15. Wu, Y. *et al.* Interplay between menin and K-Ras in regulating lung adenocarcinoma. *J Biol Chem* **287**, 40003-40011 (2012).

REVIEWERS' COMMENTS:

Reviewer #1 (Remarks to the Author):

I have no further comments

Reviewer #3 (Remarks to the Author):

NCOMMS-19-07700C-Z

The authors have provided a lot of additional data and response to the multiple comments by all of the reviewers. Overall these responses are reasonable. However, for the most important clinical translation aspect of their paper they did not perform the experiment requested. Their overall findings and conclusions are that loss of MEN1 expression leads to non small cell lung cancers with neuroendocrine differentiation and they emphasize this occurs in the setting of oncogenic KRAS mutation. They perform preclinical-targeted therapy studies and conclude that targeting LSD1 would be a new-targeted therapy for this class of NSCLCs. In fact the concluding sentence of their abstract is: " Finally, the use of LSD1 inhibitors targeting chromatin histone remodeling is expected to be an effective strategy for the treatment of lung cancer caused by Men1 deficiency. " An obvious pre requisite of bringing this strategy to the clinic would be to show in a panel of patient derived NSCLC models (cell lines and xenografts) that have low MEN1 expression KRAS mutations, that they would express neuroendocrine (NE) markers have a dramatic anti-tumor response to LSD1 inhibitors compared to NSCLCs with high MEN1 expression. Now it would also be interesting if NSCLCs with low MEN1 expression (without KRAS mutations) also express NE markers and respond to LSD1 inhibition but the key finding would be low MEN1 expression and response to LSD1 inhibition (a tumor biomarker for patient selection – low MEN1 expression) and a targeted therapy – LSD1 inhibition. Now the data they present despite my specific objection to it previously was that A549 cells, while they have a KRAS mutation, do not have low MEN1 expression or expression of NE markers. When they manipulate MEN1 with an shRNA they do find expression of NE markers and sensitivity to LSD1 inhibition. However, the very important key fact is that A549 tumor cells arose without loss of MEN1 expression. Thus, artificial manipulation of MEN1 levels does not tell us the very important fact of what happens when one studies NSCLCS that did arise with low MEN1 levels (and presumably with associated expression of NE markers). Just so the authors understand the point, NSCLCs can arise with our without an LKB1/STK11 mutation and these can behave very differently in patients. However, if you take a human NSCLC line or xenograft that is wild type for LKB1 and now artificially make it LKB1 deficient, that is not the same as a tumor that arose in a patient by virtue of having LKB1 deficiency. Thus, they failed address the single most important point for future clinical translation.